# MomentumSMoE: Integrating Momentum into Sparse Mixture of Experts

**Rachel S.Y. Teo**
Department of Mathematics
National University of Singapore
`rachel.tsy@u.nus.edu`

**Tan M. Nguyen**
Department of Mathematics
National University of Singapore
`tanmn@nus.edu.sg`

## Abstract

Sparse Mixture of Experts (SMoE) has become the key to unlocking unparalleled scalability in deep learning. SMoE has the potential to exponentially increase in parameter count while maintaining the efficiency of the model by only activating a small subset of these parameters for a given sample. However, it has been observed that SMoE suffers from unstable training and has difficulty adapting to new distributions, leading to the model's lack of robustness to data contamination. To overcome these limitations, we first establish a connection between the dynamics of the expert representations in SMoEs and gradient descent on a multi-objective optimization problem. Leveraging our framework, we then integrate momentum into SMoE and propose a new family of SMoEs, named MomentumSMoE. We theoretically prove and numerically demonstrate that MomentumSMoE is more stable and robust than SMoE. In particular, we verify the advantages of MomentumSMoE over SMoE on a variety of practical tasks including ImageNet-1K object recognition and WikiText-103 language modeling. We demonstrate the applicability of MomentumSMoE to many types of SMoE models, including those in the Sparse MoE model for vision (V-MoE) and the Generalist Language Model (GLaM). We also show that other advanced momentum-based optimization methods, such as Adam, can be easily incorporated into the MomentumSMoE framework for designing new SMoE models with even better performance, almost negligible additional computation cost, and simple implementations. The code is publicly available at https://github.com/rachtsy/MomentumSMoE.

## 1 Introduction

Scaling up deep models has demonstrated significant potential for enhancing the model's performance on a wide range of cognitive and machine learning tasks, ranging from large language model pre-training [13, 58, 59, 30, 5, 51, 70] and vision understanding [15, 2, 3, 39, 1, 40] to reinforcement learning [6, 28] and scientific applications [66, 74]. However, increasing the model's size requires a higher computational budget, which can be often challenging to meet. As a result, Sparse Mixture of Experts (SMoE) has been recently studied as an efficient approach to effectively scale up deep models. By modularizing the network and activating only subsets of experts for each input, SMoE maintains constant computational costs while increasing model complexity. This approach enables the development of billion-parameter models and achieves significant success in various applications, including machine translation [35], image classification [61], and speech recognition [34].

### 1.1 Sparse Mixture of Experts

A MoE replaces a component in the layer of the model, for example, a feed-forward or convolutional layer, by a set of networks termed experts. This approach largely scales up the model but increases

Please correspond to: `rachel.tsy@u.nus.edu`

38th Conference on Neural Information Processing Systems (NeurIPS 2024).

the computational cost. A SMoE inherits the extended model capacity from MoE but preserves the computational overhead by taking advantage of conditional computation. In particular, a SMoE consists of a router and $E$ expert networks, $u_i$, $i = 1, 2, \ldots, E$. For each input token $\boldsymbol{x}_t \in \mathbb{R}^D$ at layer $t$, the SMoE's router computes the affinity scores between $\boldsymbol{x}_t$ and each expert as $g_i(\boldsymbol{x}_t)$, $i = 1, 2, \ldots, E$. In practice, we often choose the router $g(\boldsymbol{x}_t) = [g_1(\boldsymbol{x}_t), g_2(\boldsymbol{x}_t), \ldots, g_E(\boldsymbol{x}_t)]^\top = \boldsymbol{W}\boldsymbol{x} + \boldsymbol{b}$, where $\boldsymbol{W} \in \mathbb{R}^{E \times D}$ and $\boldsymbol{b} \in \mathbb{R}^E$. Then, a sparse gating function TopK is applied to select only $K$ experts with the greatest affinity scores. Here, we define the TopK function as:

$$\mathrm{TopK}(g_i) := \begin{cases} g_i, & \text{if } g_i \text{ is in the } K \text{ largest elements of } g \\ -\infty, & \text{otherwise.} \end{cases} \tag{1}$$

The outputs from $K$ expert networks chosen by the router are then linearly combined as

$$\boldsymbol{x}_{t+1} = \boldsymbol{x}_t + \sum_{i=1}^{E} \mathrm{softmax}(\mathrm{TopK}(g_i(\boldsymbol{x}_t))u_i(\boldsymbol{x}_t) = \boldsymbol{x}_t + u(\boldsymbol{x}_t), \tag{2}$$

where $\mathrm{softmax}(g_i) := \exp(g_i)/\sum_{j=1}^{E}\exp(g_j)$. We often set $K = 2$, i.e., top-2 routing, as this configuration has been shown to provide the best trade-off between training efficiency and testing performance [35, 16, 76].

**Limitations of SMoE.** Despite their remarkable success, SMoE suffers from unstable training [11, 78] and difficulty in adapting to new distributions, leading to the model's lack of robustness to data contamination [55, 75]. These limitations impede the application of SMoE to many important large-scale tasks.

## 1.2   Contribution

In this paper, we explore the role of the residual connection in SMoE and show that simple modifications of this residual connection can help enhance the stability and robustness of SMoE. In particular, we develop a gradient descent (GD) analogy of the SMoE, showing that the dynamics of the expert representations in SMoE is associated with a gradient descent step toward the optimal solution of a multi-objective optimization problem. We then propose to integrate heavy-ball momentum into the dynamics of SMoE, which results in the Momentum Sparse Mixture-of-Experts (MomentumSMoE). At the core of MomentumSMoE is the use of momentum to stabilize and robustify the model. The architecture of MomentumSMoE is depicted in Fig. 1. MomentumSMoE can be extended beyond heavy-ball momentum to integrate well with other advanced momentum-accelerated methods such as AdamW [33, 42] and Robust Momentum [10]. Our contribution is three-fold:

1. We incorporate heavy-ball momentum in SMoE to improve the model's stability and robustness.

2. We theoretically prove that the spectrum of MomentumSMoE is better-structured than SMoE, leading to the model's stability enhancement.

3. We show that the design principle of MomentumSMoE can be generalized to other advanced momentum-based optimization methods, proposing AdamSMoE and Robust MomentumSMoE.

Our experimental results validate that our momentum-based SMoEs improve over the baseline SMoE in terms of accuracy and robustness on a variety of practical benchmarks, including WikiText-103 language modeling and ImageNet-1K object recognition. We also empirically demonstrate that our momentum-based design framework is universally applicable to many existing SMoE models, including the Sparse MoE model for vision (V-MoE) [61] and the Generalist Language Model (GLaM) [16], just by changing a few lines of the baseline SMoE code.

**Organization.** We structure this paper as follows: In Section 2, we establish the connection between SMoE and gradient descent and derive our MomentumSMoE. In Section 3, we theoretically prove the stability advantage of MomentumSMoE over SMoE. In Section 4, we introduce AdamSMoE and Robust MomentumSMoE. In Section 5, we present our experimental results to justify the advantages of our momentum-based SMoE models over the traditional SMoE and other SMoE baselines. In Section 6, we empirically analyze our MomentumSMoE. We discuss related works in Section 7. The paper ends with concluding remarks. More experimental details are provided in the Appendix.

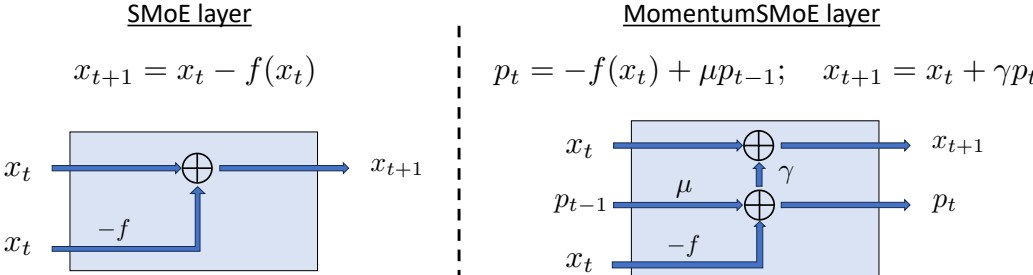

Figure 1: Illustration of SMoE (Left) and MomentumSMoE layer (Right). We establish a connection between Multiple-Gradient Descent and SMoE to introduce momentum into the model, leading to better accuracy, enhanced robustness, and faster convergence.

## 2 Momentum Sparse Mixture of Experts

### 2.1 Background: Multiple-Gradient Descent Algorithm for Multi-objective Optimization

A multi-objective optimization problem comprises of the concurrent optimization of $E$ objective functions, $F_i(x), i = 1, 2, \ldots, E$, which might be formulated as the following minimization problem

$$\min_{x \in D} F(x) := \sum_{i=1}^{E} c_i F_i(x) \tag{3}$$

where $D$ is the feasible region and $c_i \in \mathbb{R}$ are weights representing the importance of each objective function. The optimal solution to the multi-objective optimization problem above is a Pareto-optimal point such that there is no other solution that can decrease at least one of the objective functions without increasing any other objective functions. [12] shows that a necessary condition for a solution to be Pareto-optimal is for it to be Pareto-stationary, which is defined as:

**Definition 1 (Pareto-stationary)** *Let $x$ be in the interior of the feasible region, $D$, in which the $E$ objective functions, $F_i$, are smooth, and $f_i(x) = \nabla_x F_i(x)$ be the local gradients for $i = 1, \ldots, E$. $x$ is said to be Pareto-stationary if there exists a vector $\boldsymbol{\alpha} = [\alpha_1, \ldots, \alpha_E]^\top \in \mathbb{R}^E$ such that $\alpha_i \geq 0, \sum_{i=1}^{E} \alpha_i = 1$ and $\sum_{i=1}^{E} \alpha_i f_i(x) = 0$. That is, there exists a convex combination of the gradient-vectors $f_i(x)$ that is equal to $0$.*

Therefore, it would be intuitive to extend the steepest descent algorithm to a multi-objective setting by finding a descent direction, that is common to all objectives, in the convex hull of the normalized local gradients $\tilde{f}_i(x) = f_i(x)/\|f_i(x)\|$. We denote such a set as $\bar{U} = \{v \in \mathbb{R}^N | v = \sum_{i=1}^{E} \alpha_i \tilde{f}_i(x); \alpha_i \geq 0, \forall i; \sum_{i=1}^{E} \alpha_i = 1\}$. Indeed, [12] developed the Multiple-Gradient Descent Algorithm (MGDA) from such an understanding, proving that there does exist such a descent direction in $\bar{U}$, which is the direction with the smallest norm in the set. Then, the update rule of MGDA is

$$x_{t+1} = x_t - \gamma \sum_{i=1}^{E} \alpha_i^* \tilde{f}_i(x_t) \tag{4}$$

where $\alpha^* = (\alpha_1^*, \ldots, \alpha_E^*)$ minimizes $\{\|v\| | v \in \bar{U}\}$.

### 2.2 Background: Momentum Acceleration for Gradient-Based Optimization

Among the simplest learning algorithms is gradient descent, also termed the steepest descent method. It typically starts with an objective function $F(x)$ whose minima we aim to find by modifying our iterate $x_t$ at each time step $t$ through its gradient $f(x_t) = \nabla_x F(x_t)$, scaled by a step size $\gamma > 0$:

$$x_{t+1} = x_t - \gamma \nabla_x F(x_t) = x_t - \gamma f(x_t). \tag{5}$$

However, following this update rule might result in slow convergence. A classical acceleration method to speed up the steepest descent, known as the heavy-ball method [53, 67], includes a momentum term in the algorithm. This takes the form of

$$p_t = -f(x_t) + \mu p_{t-1}; \quad x_{t+1} = x_t + \gamma p_t, \tag{6}$$

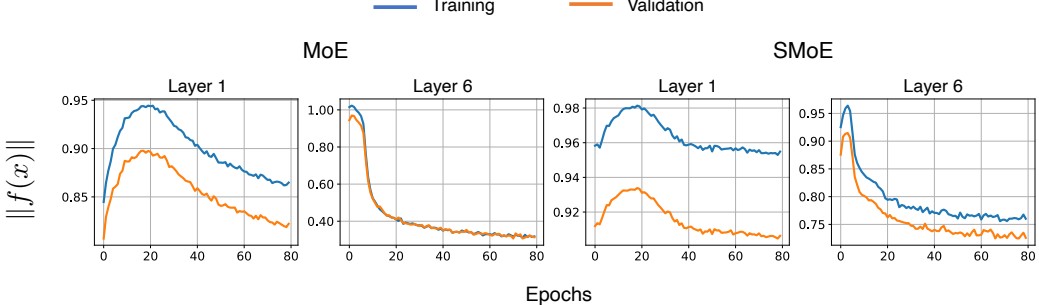

Figure 2: Average output norms at layers 1 and 6 of the MoE/SMoE during 80 training epochs on WikiText-103.

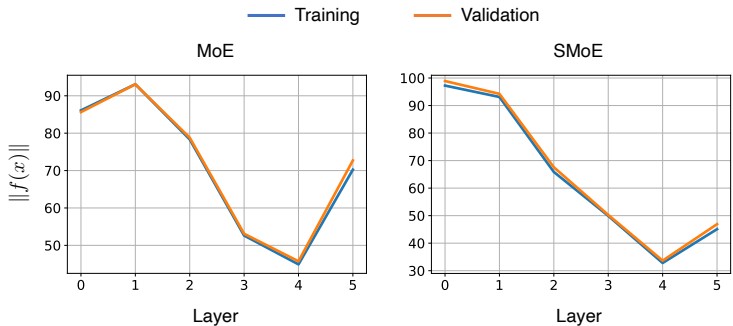

Figure 3: Average output norm at each layer across 1K train/validation samples of the (S)MoE trained on WikiText-103.

where $\gamma > 0$ is the step size, and $\mu \geq 0$ is the momentum constant. Eqn. 6 can then be rewritten as:

$$x_{t+1} = x_t + \gamma[-f(x_t) + \mu p_{t-1}] = x_t - \gamma f(x_t) + \mu(x_t - x_{t-1}). \qquad (7)$$

By incorporating the past gradients in each update, the descent path can become smoother with fewer oscillations, resulting in a faster convergence [53, 21, 71].

### 2.3 (S)MoE as (Stochastic) Gradient Descent on Multi-objective Optimization and MomentumSMoE

We will now consider the SMoE model from the multi-objective optimization perspective.

**MoE as Gradient Descent.** In viewing each expert in an SMoE as specializing in optimizing an objective function, we are going to establish a connection between MoE and GD, and further leverage momentum to enhance MoE and SMoE. We rewrite Eqn. 2 of MoE as follows:

$$\boldsymbol{x}_{t+1} = \boldsymbol{x}_t - \gamma \sum_{i=1}^{E} \text{softmax}(g_i(\boldsymbol{x}_t))[-u_i(\boldsymbol{x}_t)]. \qquad (8)$$

If we regard $-u_i(\boldsymbol{x}_t)$ as the local "gradient" $\nabla_{\boldsymbol{x}} F_i(\boldsymbol{x}_t)$ at the $t$-th iteration and $\text{softmax}(g_i(\boldsymbol{x}_t))$ as to be learned to approximate $\alpha_i^*$, then we can consider the MoE in Eqn. 2 and 8 as the dynamical system which updates $\boldsymbol{x}_t$ using the MGDA to minimize the multi-objective optimization in Eqn. 3.

**SMoE as Stochastic Gradient Descent.** Given the analogy between MoE and GD, SMoE can be interpreted as a stochastic version of MoE, which corresponds to an SGD algorithm applied to the multi-objective optimization problem in Eqn. 3. SMoE is then reformulated as:

$$\boldsymbol{x}_{t+1} = \boldsymbol{x}_t - \gamma \sum_{i=1}^{K} \text{softmax}(\text{TopK}(g_i(\boldsymbol{x}_t)))[-u_i(\boldsymbol{x}_t)] = \boldsymbol{x}_t - \gamma f(\boldsymbol{x}_t). \qquad (9)$$

Here, $-f(\boldsymbol{x}_t) = \gamma \sum_{i=1}^{K} \text{softmax}(\text{TopK}(g_i(\boldsymbol{x}_t)))u_i(\boldsymbol{x}_t)$ is the SMoE output.

**Empirical Evidences for the Gradient Descent Analogy of (S)MoE.** We provide empirical justification for the connection between (S)MoE and (S)GD in Fig. 2 and 3.

*Gradient norm $\|f(\boldsymbol{x}_t)\|$ decreases when $t$ increases:* As shown in Eqn. 8 and 9 above, the norm of the MoE and SMoE output corresponds to the gradient norm $\|f(\boldsymbol{x}_t)\| = \nabla_{\boldsymbol{x}} F(\boldsymbol{x}_t)$, respectively. It is expected that this gradient norm decreases when $t$ increases or equivalently when the number of layers in an (S)MoE model increases. Fig. 3 confirms this expectation by showing that the norm of the (S)MoE output decreases over layers in a 6-layer (S)MoE model trained on the WikiText-103 language modeling task. At the last layer, the norm increases might be due to overshooting, a common phenomenon that can occur when using gradient descent.

*Gradient norm $\|f(\boldsymbol{x}_t)\|$ at each layer $t$ decreases during training:* According to the update rule of MGDA in Eqn. 7, the coefficient $\alpha_i^*$ minimizes the norm $\|\sum_{i=1}^{E} \alpha_i \tilde{f}_i\|$. In Eqn. 8 and 9, as discussed above, $\text{softmax}(g_i(\boldsymbol{x}_t))$ and $\text{softmax}(\text{TopK}(g_i(\boldsymbol{x}_t)))$ try to learn $\alpha_i^*$, respectively. Thus, it is expected that these two terms learn to reduce the corresponding $\|\sum_{i=1}^{E} \alpha_i \tilde{f}_i\|$ in Eqn. 8 and 9, which is the norm of the SMoE output at layer $t$. Fig. 2 verifies this expectation by showing that each MoE and SMoE layer learns to reduce its output norm during training, suggesting that the routers $\text{softmax}(g_i(\boldsymbol{x}_t))$ and $\text{softmax}(\text{TopK}(g_i(\boldsymbol{x}_t)))$ learn to approximate $\alpha_i^*$. We provide the full plots for all layers in Appendix C, Fig. 6 and 7.

### 2.4 MomentumSMoE

We propose the new *MomentumSMoE* layer, depicted in Fig. 1, to accelerate the dynamics of 8, which is principled by the accelerated gradient descent theory (see Section 2.2):

$$\boldsymbol{p}_t = -f(\boldsymbol{x}_t) + \mu \boldsymbol{p}_{t-1}; \quad \boldsymbol{x}_{t+1} = \boldsymbol{x}_t + \gamma \boldsymbol{p}_t, \tag{10}$$

where $\mu \geq 0$ and $\gamma > 0$ are hyperparameters corresponding to the momentum coefficient and step size in the momentum-accelerated GD, respectively. The formulation of MomentumSMoE can be applied to MoE to derive the MomentumMoE.

## 3 Stability Analysis: MomentumSMoE vs. SMoE

In this section, we demonstrate the theoretical advantages of MomentumSMoE over SMoE. In particular, we show that the spectrum of MomentumSMoE is better-structured than that of SMoE, thus MomentumSMoE is more stable than SMoE. We rewrite MomentumSMoE using the equivalent form of momentum acceleration given in Eqn. 7 as follows:

$$\boldsymbol{x}_{t+1} = \boldsymbol{x}_t - \gamma f(\boldsymbol{x}_t) + \mu(\boldsymbol{x}_t - \boldsymbol{x}_{t-1}). \tag{11}$$

Taking inspiration from [57], we then expand $f(\boldsymbol{x}_t)$ around the Pareto-stationary solution $\boldsymbol{x}^*$ at which $f(\boldsymbol{x}^*) = 0$ (see Definition 1) using Taylor expansion to obtain an approximation of $f(\boldsymbol{x}_t)$:

$$f(\boldsymbol{x}_t) \approx f(\boldsymbol{x}^*) + \nabla_{\boldsymbol{x}} f(\boldsymbol{x}^*)(\boldsymbol{x}_t - \boldsymbol{x}^*) = \nabla_{\boldsymbol{x}} f(\boldsymbol{x}^*)(\boldsymbol{x}_t - \boldsymbol{x}^*). \tag{12}$$

Substituting the Taylor expansion of $f(\boldsymbol{x}_t)$ in Eqn. 12 into Eqn. 11, we attain

$$\boldsymbol{x}_{t+1} = \boldsymbol{x}_t - \gamma \nabla_{\boldsymbol{x}} f(\boldsymbol{x}^*)(\boldsymbol{x}_t - \boldsymbol{x}^*) + \mu(\boldsymbol{x}_t - \boldsymbol{x}_{t-1}).$$

Without loss of generality, we further let $\boldsymbol{x}^* = 0$ as we can always replace $\boldsymbol{x}_t$ with $\boldsymbol{x}_t + \boldsymbol{x}^*$. The formula of MomentumSMoE can then be simplified as

$$\boldsymbol{x}_{t+1} = \boldsymbol{x}_t - \gamma \nabla_{\boldsymbol{x}} f(\boldsymbol{x}^*)\boldsymbol{x}_t + \mu(\boldsymbol{x}_t - \boldsymbol{x}_{t-1}). \tag{13}$$

Suppose that $\nabla_{\boldsymbol{x}} f(\boldsymbol{x}^*)$ does not have any defective eigenvalues and hence is diagonalizable. Then, $\nabla_{\boldsymbol{x}} f(\boldsymbol{x}^*) = \boldsymbol{Q} \boldsymbol{\Sigma} \boldsymbol{Q}^{-1}$, for some invertible matrix $\boldsymbol{Q}$ and the diagonal matrix $\boldsymbol{\Sigma}$ with diagonal entries being the eigenvalues $\sigma(n)$, $n = 1, 2, \ldots, N$, of $\nabla_{\boldsymbol{x}} f(\boldsymbol{x}^*)$. We can then rewrite Eqn. 13 as

$$\boldsymbol{x}_{t+1} = \boldsymbol{x}_t - \gamma \boldsymbol{\Sigma} \boldsymbol{x}_t + \mu(\boldsymbol{x}_t - \boldsymbol{x}_{t-1}). \tag{14}$$

Since we have decoupled the $N$ features in $\boldsymbol{x}_t$, we can consider each feature $\boldsymbol{x}_t(n)$, separately. Introducing a dummy equation $\boldsymbol{x}_t = \boldsymbol{x}_t$, we rewrite Eqn. 14 as follows:

$$\begin{pmatrix} \boldsymbol{x}_t(n) \\ \boldsymbol{x}_{t+1}(n) \end{pmatrix} = \begin{pmatrix} 0 & 1 \\ -\mu & (1+\mu) - \gamma\sigma(n) \end{pmatrix} \begin{pmatrix} \boldsymbol{x}_{t-1}(n) \\ \boldsymbol{x}_t(n) \end{pmatrix} = \boldsymbol{A} \begin{pmatrix} \boldsymbol{x}_{t-1}(n) \\ \boldsymbol{x}_t(n) \end{pmatrix} \tag{15}$$

The convergence of $\boldsymbol{x}_t(n)$ then depends on the eigenvalues $\lambda_1(\boldsymbol{A})$ and $\lambda_2(\boldsymbol{A})$ of $\boldsymbol{A}$. In particular, we require $\max\{|\lambda_1(\boldsymbol{A})|, |\lambda_2(\boldsymbol{A})|\} < 1$. It should be noted that omitting the momentum parameter, i.e., $\mu = 0$, recovers the standard, unaccelerated SMoE layer.

**Lemma 1** *Given the matrix $\boldsymbol{A} = \begin{pmatrix} 0 & 1 \\ -\mu & (1+\mu) - \gamma\sigma(n) \end{pmatrix}$ and $\lambda_1(\boldsymbol{A})$, $\lambda_2(\boldsymbol{A})$ are eigenvalues of A, $\max\{|\lambda_1(\boldsymbol{A})|, |\lambda_2(\boldsymbol{A})|\} < 1$ if and only if $\mu \in (-1,1)$ and $\gamma\sigma(n) \in (0, 2+2\mu)$.*

**Proposition 1 (Convergence of MomentumSMoE)** *The MomentumSMoE defined in Eqn. 10 converges if and only if $\mu \in (-1,1)$ and $\gamma\sigma(n) \in (0, 2+2\mu)$.*

The proofs of the results above are provided in Appendix A. It is worth noting that in both the MomentumSMoE and standard SMoE, the convergence of $\boldsymbol{x}_t$ depends on the eigenvalues of the Jacobian $\nabla_{\boldsymbol{x}} f$ of the SMoE layer. Since the step size $\gamma > 0$, we require that $\nabla_{\boldsymbol{x}} f$ to be positive definite for its eigenvalues $\sigma(n)$ to be positive. Furthermore, even though among the convergence conditions of MomentumSMoE is that $\mu \in (-1,1)$, in practice, $\mu$ is chosen to be positive.

Proposition 1 implies that the spectrum of MomentumSMoE is better-structured than that of SMoE. Thus, MomentumSMoE is more stable than SMoE. We summarize this finding in Corollary 1 below.

**Corollary 1 (MomentumSMoE is more stable than SMoE)** *Without momentum, $\mu = 0$, the range of values that $\gamma\sigma(n)$ can take for the system to be stable is limited to $0 < \gamma\sigma(n) < 2$. The addition of momentum expands this margin, almost doubling it, providing a larger parameter range for the network to converge stably to a good output in the forward pass.*

## 4 Beyond Heavy-ball Momentum: AdamSMoE and Robust MomentumSMoE

In addition to heavy-ball momentum, there are several advanced momentum-based algorithms in optimization that can be utilized for SMoE design. In this subsection, we propose two additional variants of MomentumSMoE, AdamSMoE and Robust MomentumSMoE, which are derived from the AdamW [33, 42] and Robust Momentum [10], respectively.

### 4.1 Adam Sparse Mixture of Experts (AdamSMoE)

Adam [33] accelerates the gradient dynamics by utilizing the moving average of historical gradients and element-wise squared gradients. Adam with a decoupled weight decay regularization (AdamW) is more commonly used in practice thanks to its better generalization over Adam. We employ AdamW to derive the *AdamSMoE* as follows:

$$\boldsymbol{p}_t = \mu\boldsymbol{p}_{t-1} + (1-\mu)[-f(\boldsymbol{x}_t)]; \quad \boldsymbol{m}_t = \beta\boldsymbol{m}_{t-1} + (1-\beta)f(\boldsymbol{x}_t) \odot f(\boldsymbol{x}_t)$$

$$\boldsymbol{x}_{t+1} = \boldsymbol{x}_t + \frac{\gamma}{\sqrt{\boldsymbol{m}_t} + \epsilon}\boldsymbol{p}_t - \kappa\boldsymbol{x}_t$$

where $\epsilon$ is a small constant to prevent numerical instability, $\kappa$ the weight decay parameter, $\gamma$ the step size, and $\mu$ and $\beta$ are the decay parameters for the moment estimates.

### 4.2 Robust Momentum Sparse Mixture of Experts (Robust MomentumSMoE)

Deep learning models, including SMoE, are known to not be robust to distribution shifts and data distortions [60, 19, 14]. Utilizing the connection between (S)MoE and (S)GD in Section 2.3, we develop the new *Robust MomentumSMoE* from the Robust Momentum Method [10].

The Robust Momentum Method proposed by [10] has the following update rule

$$y_t = x_t + \alpha(x_t - x_{t-1}); \quad x_{t+1} = x_t - \gamma f(y_t) + \mu(x_t - x_{t-1}), \tag{16}$$

where $\gamma$, $\mu$ and $\alpha$ are parameterized by an additional hyperparameter $p$ as follows:

$$\gamma = \frac{k(1-p)^2(1+p)}{L}; \quad \mu = \frac{kp^3}{k-1}; \quad \alpha = \frac{p^3}{(k-1)(1-p)^2(1+p)}. \tag{17}$$

Here, $k = L/m$ is a condition ratio of the objective function assuming that it is $m$-strongly convex and $L$-smooth. Compared with the heavy-ball momentum in Eqn. 7, there is an additional variable $y_t$ that can be interpreted as a feedback signal to steer the $x_t$ toward a robust solution. The parameters $\gamma$, $\mu$, and $\alpha$ are designed such that the new system is robust.

Table 1: Perplexity (PPL) of momentum-based SMoE vs. SMoE baseline on clean/attacked WikiText-103.

| Model/Metric | Parameters | Clean WikiText-103 | | Attacked WikiText-103 | |
|---|---|---|---|---|---|
| | | Valid PPL ↓ | Test PPL ↓ | Valid PPL ↓ | Test PPL ↓ |
| *SMoE-medium (baseline)* | 216M | 33.76 | 35.55 | 42.24 | 44.19 |
| MomentumSMoE-medium | 216M | 32.29 | 33.46 | 40.94 | 42.33 |
| AdamSMoE-medium | 216M | **31.59** | **33.25** | **39.27** | **41.11** |
| *SMoE-large (baseline)* | 388M | 29.31 | 30.33 | 36.77 | 37.83 |
| MomentumSMoE-large | 388M | **27.58** | **29.03** | **35.21** | **36.78** |
| *GLaM-medium (baseline)* | 220M | 36.37 | 37.71 | 45.83 | 47.61 |
| MomentumGLaM-medium | 220M | 33.87 | 35.29 | 42.15 | 43.64 |
| AdamGLaM-medium | 220M | **32.99** | **34.32** | **41.09** | **42.81** |

We incorporate the Robust Momentum Method above in our SMoE optimization framework developed in Section 2.3 and formulate the novel Robust MomentumSMoE as follows:

$$\boldsymbol{y}_t = \boldsymbol{x}_t + \alpha(\boldsymbol{x}_t - \boldsymbol{x}_{t-1}); \quad \boldsymbol{x}_{t+1} = \boldsymbol{x}_t - \gamma f(\boldsymbol{y}_t) + \mu(\boldsymbol{x}_t - \boldsymbol{x}_{t-1}), \tag{18}$$

where $\gamma$, $\mu$ and $\alpha$ are as defined in Eqn. 17, and $-f(\boldsymbol{y}_t) = \gamma \sum_{i=1}^{K} \text{softmax}(\text{TopK}(g_i(\boldsymbol{y}_t)))u_i(\boldsymbol{y}_t)$ is the SMoE output given the input $\boldsymbol{y}_t$. Equivalently, at each layer $t$, we update the input $\boldsymbol{x}_t$ and momentum vector $\boldsymbol{p}_t$ as

$$\boldsymbol{y}_t = \boldsymbol{x}_t + \alpha\gamma\boldsymbol{p}_{t-1}; \quad \boldsymbol{p}_t = -f(\boldsymbol{y}_t) + \mu\boldsymbol{p}_{t-1}; \quad \boldsymbol{x}_{t+1} = \boldsymbol{x}_t + \gamma\boldsymbol{p}_t.$$

**Remark 1** *We provide an interpretation of robust momentum in Appendix B.*

## 5 Experimental Results

In this section, we numerically justify the advantages of our momentum-based SMoE over the baseline SMoE on both WikiText-103 language modeling and ImageNet-1k object recognition tasks. We aim to show that: (i) MomentumSMoE improves model performance across both language and vision tasks; (ii) AdamSMoE significantly outperforms the baseline and accelerates convergence in language models, even surpassing MomentumSMoE; (iii) Robust MomentumSMoE is highly effective at improving robustness of vision models to data corruption; (iv) MomentumSMoE is universal and can be easily integrated into many state-of-the-art SMoE and MoE models.

Throughout the experiments, we compare our momentum-based SMoE with the baseline SMoE of the same configuration, replacing SMoE layers with our momentum-based SMoE. For Adam-based SMoE models, we use AdamSMoE in the first layer of the model and MomentumSMoE for the subsequent layers. We provide an explanation for this implementation in Appendix D.1. We find that implementing AdamSMoE in the first layer is enough to significantly improve the model's overall performance and accelerate its convergence. Our results are averaged over 5 runs. Details on datasets, models, and training are provided in Appendix D, along with Table 4 summarizing the momentum methods implemented on SMoE/MoE models for different tasks in our experiments. More results can be found in Appendix E. All experiments are conducted on a server with 8 A100 GPUs.

### 5.1 WikiText-103 Language Modeling

We use the Switch Transformer [18], referred to as SMoE in our tables and figures below, and GLaM [16] baselines. We consider 2 configurations: medium (6 layers) and large (12 layers). We report the perplexity (PPL) of MomentumSMoE and AdamSMoE in comparison with the baseline SMoE on word-level WikiText-103 validation and test datasets for both model sizes in Table 1. We also include experiments on the medium-sized GLaM. A lower PPL indicates a better performance of the model. To further demonstrate the robustness of our method, we test the models on word swap attacked WikiText-103 data and present their results. Across all metrics, AdamSMoE and AdamGLaM outperform the baseline by a significant margin, verifying the strength of our method. Additionally, in Figure 4(Left), we provide the training and validation PPL during the first 5 training epochs of MomentumSMoE and AdamSMoE compared to the baseline SMoE to illustrate the accelerated convergence of our momentum-based models.

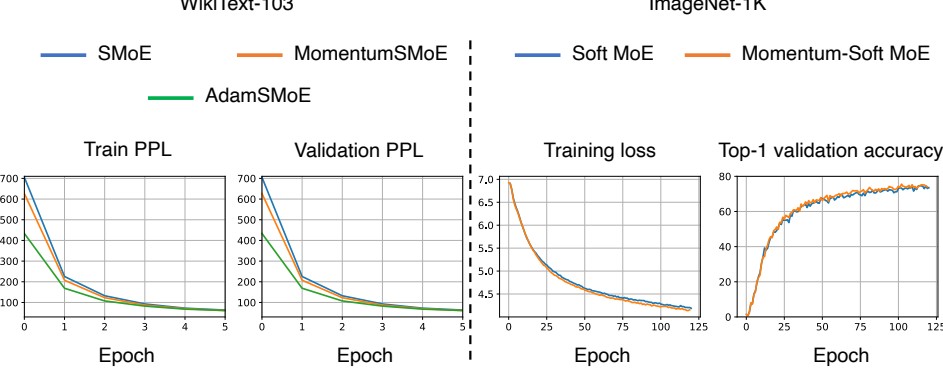

Figure 4: **Left:** WikiText-103 train/validation perplexity (PPL) curves during the first 5 training epochs for MomentumSMoE, AdamSMoE, and SMoE. AdamSMoE has significantly faster convergence compared to SMoE. **Right:** Training loss/top-1 accuracy (%) of Momentum-Soft MoE vs. Soft MoE baseline on ImageNet-1K across 120 epochs of training. Momentum-Soft MoE has faster convergence and improved accuracy.

Table 2: Top-1 accuracy (%) and mean corruption error (mCE) of MomentumV-MoE and Robust MomentumV-MoE vs. the V-MoE baseline on ImageNet-1K and popular robustness benchmarks for image classification.

| Model | Params | Train IN-1K | | Valid IN-1K | | IN-R | IN-A | IN-C | |
|---|---|---|---|---|---|---|---|---|---|
| | | Top-1 ↑ | Top-5 ↑ | Top-1 ↑ | Top-5 ↑ | Top-1 ↑ | Top-1 ↑ | Top-1 ↑ | mCE ↓ |
| *V-MoE (baseline)* | 297M | 76.49 | **92.27** | 73.16 | **90.42** | 36.10 | 5.25 | 46.98 | 67.14 |
| MomentumV-MoE | 297M | **76.92** | 92.19 | **73.26** | 90.30 | 37.45 | **6.48** | 48.11 | 65.77 |
| Robust MomentumV-MoE | 297M | 76.66 | **92.27** | 73.20 | 90.36 | **37.57** | 6.37 | **48.82** | **64.92** |

An important advantage of MomentumSMoE is its simplicity, which allows easy implementation with negligible computational overhead. We provide a comparison of the run time/sample, memory, number of parameters, and computation time between models in Table 11 and 12 in Appendix E.5.

## 5.2 ImageNet-1K Object Recognition Task

In this section, we investigate our momentum-based models on two popular vision models, Vision MoE (V-MoE) [61] and Soft MoE [56] on the ImagetNet-1K (IN-1K) object recognition task. We focus on $i$) improving the robustness of V-MoE using Robust MomentumSMoE (18) and $ii$) demonstrating that our momentum method is not limited to sparse models but can be generalized to MoE models such as Soft MoE. To benchmark robustness to data corruptions, we use the standard datasets, ImageNet-R (IN-R) [24], ImageNet-A (IN-A) [26], and ImageNet-C (IN-C) [25].

**Vision Mixture of Experts (V-MoE).** We use a V-MoE (small) model as the baseline. This V-MoE consists of 8 Vision Transformer (ViT) blocks [15] with every odd block's MLP being replaced by a SMoE layer. In Table 2, we provide the top-1 accuracy (%) on the training and validation set of IN-1K, IN-R, IN-A, and IN-C, as well as the mean Corruption Error (mCE) for IN-C. While MomentumV-MoE and Robust MomentumV-MoE have marginal gains on clean IN-1K data, we see significant improvement on IN-R, IN-A, and IN-C with at least a 1% increase in accuracy across these metrics. Specifically, Robust MomentumV-MoE has an almost 2% increase and 2 mCE decrease on IN-C, justifying the advantage of our method. Furthermore, we visualize the top-1 accuracy and mCE across increasing severity of two corruption types in Fig. 14 in Appendix E.3 to illustrate the increasing effectiveness of our method with escalating data corruption. The results of Robust MomentumSMoE on WikiText-103 can also be found in Appendix E.4, Table 9.

**Soft Mixture of Experts (Soft MoE).** We use the Soft MoE-tiny with 12 layers. The first 6 layers consist of standard ViT blocks, and the last 6 layers replace the MLP in those blocks with a Soft MoE layer. We train a Momentum-Soft MoE and a baseline Soft MoE model on ImageNet-1K and present their results in Table 3. In addition, we plot the training loss and top-1 accuracy of both models for 120 training epochs in Fig. 4(Right). Notably, there is a considerable increase in the accuracy of Momentum-Soft MoE over the baseline, as well as a clear acceleration to a good solution during

Table 3: Top-1/top-5 accuracy (%) of Momentum-Soft MoE vs. Soft MoE baseline on ImageNet-1K (IN-1K).

| Model | Params | Valid IN-1K | |
| | | Top-1 ↑ | Top-5 ↑ |
| --- | --- | --- | --- |
| *Soft MoE (baseline)* | 231M | 73.52 | 90.94 |
| Momentum-Soft MoE | 231M | **75.47** | **92.34** |

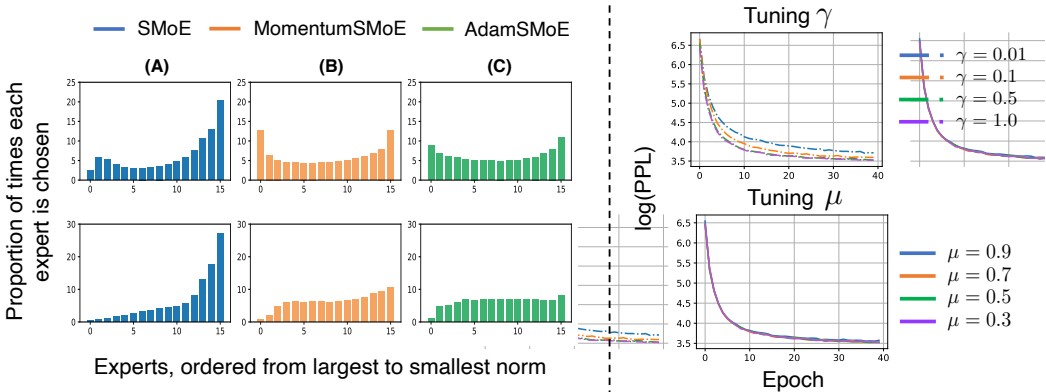

Figure 5: **Left:** Proportion of each expert chosen, ordered from the largest norm of each expert output to the smallest norm, in layers 3 and 5 of SMoE, MomentumSMoE, and Adam SMoE, averaged over the WikiText-103 validation set. **Right:** Log validation perplexity (PPL) during the finetuning of hyperparameters, $\mu$ and $\gamma$, for 40 training epochs in MomentumSMoE. When tuning $\gamma$, we keep $\mu = 0.7$ and vice versa with $\gamma = 1.0$.

training. These findings justify the benefits and universality of our momentum-based method, that extends beyond SMoE to MoE.

## 6 Empirical Analysis

We conduct empirical analysis based on the SMoE-medium trained on WikiText-103 in Section 5.1.

**Norm-based Load Imbalance.** Load imbalance in SMoE occurs when only a small subset of experts are consistently selected [63, 77]. Numerous methods have been developed to counter this common phenomenon, such as introducing a buffer capacity for each expert and a load balancing loss [65, 18]. Orthogonal to these, in line with our GD framework for SMoEs, we examine the choice of experts determined by the size of the norm of their outputs, $\|f_i(x)\|$.

From a multi-objective optimization perspective, the optimal descent direction is one that minimizes the norm in the convex hull of the normalized gradients $\bar{U}$ (see Section 2.1). If the gradients are not normalized, the minimum norm direction is then expected to be mainly influenced by the gradients with the smallest norms [12]. From our GD analogy of SMoE in Section 2.3, the gradients correspond to the experts, whose outputs are not normalized. We then visualize the proportion of times the experts in a SMoE are chosen according to their norms during inference in Fig. 5(Left, A). We exactly observe the corresponding phenomenon in the SMoE, further empirically justifying our connection between SMoE and GD. The full plots for all layers and all models are provided in Appendix C.

The direction with the smallest norms are frequently related to the objectives that have already had a substantial degree of convergence and is insufficient for a balanced minimization of the multi-objective criteria. In this light, an ideal SMoE output would have a norm-based balanced choice of experts and should translate to improved model performance. Indeed, in Section 5.1, we established the superior performance of MomentumSMoE and AdamSMoE on large-scale WikiText-103 language modeling task, and in Figure 5(Left, B, C), this directly correlates with a significantly more balanced selection of experts with respect to their norms.

**Ablation Study on Momentum $\mu$ and Step Size $\gamma$.** To better understand the effects of the momentum parameter and step size on the performance of the trained MomentumSMoE models, we do an ablation

study on these two hyperparameters and include results in Fig. 5(Right), which contains a plot of log validation PPL during 40 training epochs. We notice that MomentumSMoE is robust to the choice of $\mu$, and we select $\mu = 0.7$ for the final comparison with the baseline SMoE. On the other hand, when the value of $\gamma$ is too small, there is an adverse effect on the model. Hence, we select $\gamma = 1.0$.

**Making Momentum $\mu$ and Step Size $\gamma$ Learnable.** We note that additional time and effort are required to tune the momentum parameter and step size. Thus, we explore different methods to circumvent this limitation. We discuss the results of one such method, making $\gamma$ and $\mu$ learnable parameters, in this section and include results in Table 5 in Appendix E.1. We leave the discussion on other methods to Appendix E.1. As shown in Table 5 in Appendix E.1, MomentumSMoE $(ii)$, with learnable $\gamma$ and fixed $\mu$, significantly outperforms the baseline for both clean validation and test data by at least 1.5 PPL, even surpassing the tuned model in Table 1. On the attacked data, the benefits of MomentumSMoE are further enhanced with more than 2 PPL improvements. These results confirm that the benefits of our model can be leveraged with minimal effort. Since the MomentumSMoE is robust to the choice of $\mu$, we do not consider the setting of fixing $\gamma$ and making $\mu$ learnable.

**Other Optimizers.** We study the integration of other advanced momentum and optimization methods, such as the Nesterov accelerated gradient [48], RMSProp [68], and sharpness-aware minimization [64], into our MomentumSMoE framework in Appendix E.4.

# 7 Related Work

**Sparse Mixture of Experts.** SMoE has been extensively studied to enhance the training efficiency of large language models (LLMs), with various stable routing strategies proposed, including (i) allowing tokens to select the top-k experts [35, 18, 79, 11], (ii) allowing experts to select the top-k tokens [77], and (iii) globally determining expert assignment [36, 9]. Recent works have also tried to enhance the robustness of SMoE. [55] study the robustness of SMoE for ViTs [15] while [75] investigates the robustness of SMoE for CNNs. Furthemore, [37] explores the potential of SMoE for domain generalization, and [22] employs SMoE for robust multi-task learning. Various works have also focused on addressing load imbalance in SMoE, including [63, 77, 7]. Our momentum-based SMoE can be easily incorporated into these methods above to further improve their performance.

**Deep Learning Models with Momentum.** Momentum has been utilized in the design of deep neural network (DNN) architectures [72, 38]. [23] applies momentum to create large and consistent dictionaries for unsupervised learning using a contrastive loss, with a momentum-based moving average of the queue encoder at the core of this approach. Many DNN-based methods for sparse coding have been designed by unfolding classical optimization algorithms, such as FISTA [4], where momentum is used in the underlying optimizer [45]. In addition, [38] introduces momentum into ResNet and DenseNet, [73, 49] integrate momentum into neural differential equations, [52] incorporates momentum into transformers, and [50] designs RNNs using momentum-accelerated first-order optimization algorithms.

# 8 Concluding Remarks

In this paper, we propose MomentumSMoE, a new class of SMoE that utilizes heavy-ball momentum to stabilize and robustify SMoE. We theoretically justify the stability of our MomentumSMoE models compared to the SMoE baseline. Furthermore, we demonstrate that our momentum-based design framework for SMoE is universal and can incorporate advanced momentum-based optimization methods, including AdamW [33, 42] and Robust Momentum [10], into many existing SMoE models. We empirically validate the advantage of our momentum-based SMoE over the standard SMoE baseline on WikiText-103 and ImageNet-1K. As shown in Table 7 in Appendix E.2, momentum has no positive effect on the small SMoE of only 3 layers but attains an increasing improvement with the medium and large models of 6 and 12 layers, respectively. This is expected as each layer represents an iteration of GD. A limitation of MomentumSMoE is that while beneficial for larger models, for models that have few layers, MomentumSMoE has little effect. From a theoretical perspective, it would be intriguing to develop a theory to explain the enhanced robustness of MomentumSMoE. Furthermore, MomentumSMoE can be analyzed as a fixed-point iteration. We leave these theoretical developments as future work.

## Acknowledgments and Disclosure of Funding

This research / project is supported by the National Research Foundation Singapore under the AI Singapore Programme (AISG Award No: AISG2-TC-2023-012-SGIL). This research / project is supported by the Ministry of Education, Singapore, under the Academic Research Fund Tier 1 (FY2023) (A-8002040-00-00, A-8002039-00-00). This research / project is also supported by the NUS Presidential Young Professorship Award (A-0009807-01-00).

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

# Supplement to "MomentumSMoE: Integrating Momentum into Sparse Mixture of Experts"

**Table of Contents**

## A  Proof of Lemma 1

We can find the eigenvalues of matrix $A$ explicitly as

$$\lambda_{\{\substack{1 \\ 2}\}} = \frac{(1 + \mu - \gamma\sigma(n)) \pm \sqrt{(1 + \mu - \gamma\sigma(n))^2 - 4\mu}}{2}$$

If $\gamma\sigma(n) \leq 0$,

$$
\begin{aligned}
\lambda_1 &= \frac{(1 + \mu - \gamma\sigma(n)) + \sqrt{(1 + \mu - \gamma\sigma(n))^2 - 4\mu}}{2} \\
&\geq \frac{(1 + \mu) + \sqrt{(1 + \mu)^2 - 4\mu}}{2} \\
&= \frac{(1 + \mu) + |1 - \mu|}{2} \\
&= \max\{1, \mu\} \\
&\geq 1
\end{aligned}
$$

Then, we must have $\mu\sigma(n) > 0$. Letting the expression in the square root be $\Delta$, expanding it,

$$
\begin{aligned}
\Delta &:= (1 + \mu - \gamma\sigma(n))^2 - 4\mu \\
&= (1 - \gamma\sigma(n))^2 + 2(1 - \gamma\sigma(n))\mu + \mu^2 - 4\mu \\
&= \mu^2 - 2(1 - \gamma\sigma(n))\mu + (1 - \gamma\sigma(n))^2
\end{aligned}
$$

and finding the roots of the equation yields, $\mu = (1 \pm \sqrt{\gamma\sigma(n)})^2$. Then, we consider two cases.

**Case 1:** $\Delta \geq 0$ when $\mu \geq (1 + \sqrt{\gamma\sigma(n)})^2$ or $\mu \leq (1 - \sqrt{\gamma\sigma(n)})^2$

**1a:** If $\mu \geq (1+\sqrt{\gamma\sigma(n)})^2$, then $1+\mu-\gamma\sigma(n) \geq 1+(1+\sqrt{\gamma\sigma(n)})^2-\gamma\sigma(n) \geq 2+2\sqrt{\gamma\sigma(n)} \geq 0$. Then, $\lambda_{1,2} > 0$ and $\lambda_1 \geq \lambda_2$. For the system to converge, we need

$$\lambda_1 = \frac{(1 + \mu - \gamma\sigma(n)) + \sqrt{\Delta}}{2} < 1 \iff \sqrt{\Delta} < 1 - \mu + \gamma\sigma(n) \qquad (19)$$

However, by assumption, we have a contradiction

$$0 \leq \sqrt{\Delta} < 1 - \mu + \gamma\sigma(n) \leq 1 - (1 + \sqrt{\gamma\sigma(n)})^2 + \gamma\sigma(n) = -2\sqrt{\gamma\sigma(n)} < 0.$$

Hence, we cannot have $\mu \geq (1 + \sqrt{\gamma\sigma(n)})^2$.

**1b:** If $\mu \leq (1 - \sqrt{\gamma\sigma(n)})^2$, we further divide into two more subcases,

    **1bi:** If $1 + \mu - \gamma\sigma(n) \geq 0$, again we have, $\lambda_{1,2} > 0$ and $\lambda_1 \geq \lambda_2$. By Eqn. 19, we require $0 \leq \sqrt{\Delta} < 1-\mu+\gamma\sigma(n)$. As $1-\mu+\gamma\sigma(n) \geq 1-(1-\sqrt{\gamma\sigma(n)})^2+\gamma\sigma(n) = 2\sqrt{\gamma\sigma(n)} > 0$, it is enough to check that $\Delta < (1 - \mu + \gamma\sigma(n))^2$ can be satisfied.

$$
\begin{aligned}
\Delta < (1 - \mu + \gamma\sigma(n))^2 &\iff (1 + \mu - \gamma\sigma(n))^2 - 4\mu < (1 - \mu + \gamma\sigma(n))^2 \\
&\iff (1 + \mu - \gamma\sigma(n))^2 - (1 - \mu + \gamma\sigma(n))^2 < 4\mu \\
&\iff (1 + \mu - \gamma\sigma(n) + 1 - \mu + \gamma\sigma(n)) \\
&\qquad\qquad (1 + \mu - \gamma\sigma(n) - 1 + \mu - \gamma\sigma(n)) < 4\mu \\
&\iff 4(\mu - \gamma\sigma(n)) < 4\mu \\
&\iff 0 < \gamma\sigma(n)
\end{aligned}
$$

Therefore, the conditions for convergence are $i)$ $\mu \leq (1 - \sqrt{\gamma\sigma(n)})^2$, $ii)$ $1 + \mu - \gamma\sigma(n) \geq 0$ and $iii)$ $\gamma\sigma(n) > 0$. Then by $ii)$, $\mu \geq \gamma\sigma(n) - 1 \geq -1$. Suppose for a contradiction that $\mu \geq 1$, then by $i)$, $1 \leq \mu \leq (1 - \sqrt{\gamma\sigma(n)})^2$ which implies that $\gamma\sigma(n) \geq 2\sqrt{\gamma\sigma(n)}$ and hence, $\gamma\sigma(n) \geq 4$. Combining this with condition $i)$, we have $\sqrt{\mu} \leq \sqrt{\gamma\sigma(n)} - 1$ which leads to the following contradiction with $ii)$

$$
\begin{aligned}
\mu \leq (1 - \sqrt{\gamma\sigma(n)})^2 &\implies \mu \leq 1 - 2\sqrt{\gamma\sigma(n)} + \gamma\sigma(n) \\
&\implies \mu + 2(\sqrt{\gamma\sigma(n)} - 1) \leq \gamma\sigma(n) - 1 \\
&\implies \mu + 2\sqrt{\mu} \leq \gamma\sigma(n) - 1 \\
&\implies 1 + \mu - \gamma\sigma(n) + 2\sqrt{\mu} \leq 0.
\end{aligned}
$$

Then, we must have a last condition for convergence, $iv)$ $|\mu| < 1$.

    **1bii:** If $1 + \mu - \gamma\sigma(n) \leq 0$, then $|\lambda_2| \geq |\lambda_1|$ and

$$\lambda_2 = \frac{(1 + \mu - \gamma\sigma(n)) - \sqrt{\Delta}}{2} < 0$$

For the system to be stable, we need $\lambda_2 > -1 \iff 3 + \mu - \gamma\sigma(n) > \sqrt{\Delta} \geq 0$. Hence, we require $i)$ $\mu \leq (1 - \sqrt{\gamma\sigma(n)})^2$, $ii)$ $1 + \mu - \gamma\sigma(n) \leq 0$, $iii)$ $3 + \mu - \gamma\sigma(n) > 0$ and $iv)$

$$
\begin{aligned}
(3 + \mu - \gamma\sigma(n))^2 > \Delta &\iff (3 + \mu - \gamma\sigma(n))^2 > (1 + \mu - \gamma\sigma(n))^2 - 4\mu \\
&\iff 4\mu > (1 + \mu - \gamma\sigma(n))^2 - (3 + \mu - \gamma\sigma(n))^2 \\
&\iff 4\mu > (1 + \mu - \gamma\sigma(n) + 3 \qquad\qquad\qquad + \mu - \gamma\sigma(n)) \\
&\qquad\qquad (1 + \mu - \gamma\sigma(n) - 3 - \mu + \gamma\sigma(n)) \\
&\iff 4\mu > -4(2 + \mu - \gamma\sigma(n)) \\
&\iff 2 + 2\mu - \gamma\sigma(n) > 0
\end{aligned}
$$

Combining $ii)$ and $iv)$, we have, $2+2\mu-\mu-1 > 0 \implies \mu > -1$. Similarly, we assume for a contradiction that $\mu \geq 1$. Then, $\gamma\sigma(n) \geq 1+\mu \geq 2$ and by condition $i)$, $\sqrt{\mu} \leq \sqrt{\gamma\sigma(n)} - 1$

from which follows the contradiction

$$1 \leq \sqrt{\gamma\sigma(n)} - \sqrt{\mu} \implies \sqrt{\gamma\sigma(n)} + \sqrt{\mu} \leq (\sqrt{\gamma\sigma(n)} - \sqrt{\mu})(\sqrt{\gamma\sigma(n)} + \sqrt{\mu}) = \gamma\sigma(n) - \mu$$
$$\implies \sqrt{\gamma\sigma(n)} + \sqrt{\mu} \leq \gamma\sigma(n) - \mu < 3 \quad \text{(by } iii))$$
$$\implies \sqrt{\mu} + \sqrt{\gamma\sigma(n)} - 1 < 2$$
$$\implies 2\sqrt{\mu} < 2 \quad \text{(by } i) \text{ again)}.$$

Therefore, we need $v)$ $|\mu| < 1$ for convergence as well.

To summarize, for **Case 1**, in order to have a stable system, we require:

$a)$    $\mu \leq (1 - \sqrt{\gamma\sigma(n)})^2$

$b)$    If $1 + \mu - \gamma\sigma(n) \geq 0$,

      $i)$    $\gamma\sigma(n) > 0$
      $ii)$   $|\mu| < 1$

$c)$    If $1 + \mu - \gamma\sigma(n) \leq 0$,

      $i)$   $2 + 2\mu - \gamma\sigma(n) > 0$     $\implies$  
$\begin{cases} i) & 3 + \mu - \gamma\sigma(n) > 0 \\ ii) & 2 + 2\mu - \gamma\sigma(n) > 0 \\ iii) & |\mu| < 1 \end{cases}$
      $ii)$   $|\mu| < 1$

**Case 2:**   $\Delta < 0$ when $(1 + \sqrt{\gamma\sigma(n)})^2 > \mu > (1 - \sqrt{\gamma\sigma(n)})^2$

Then, the eigenvalues are complex and given by

$$\lambda_{\{\frac{1}{2}\}} = \frac{(1 + \mu - \gamma\sigma(n)) \pm i\sqrt{4\mu - (1 + \mu - \gamma\sigma(n))^2}}{2}.$$

For convergence, we require $|\lambda_{1,2}| = \sqrt{\mu} < 1$ or equivalently, $\mu < 1$. Hence, the necessary condition becomes $1 > \mu > (1 - \sqrt{\gamma\sigma(n)})^2$ and to avoid a contradiction, we also require $1 > (1 - \sqrt{\gamma\sigma(n)})^2$. This is satisfied when $\gamma\sigma(n) < 2 + 2\mu$ and $|\mu| < 1$.

Therefore, from all the considered cases, to have a stable system, we require $0 < \gamma\sigma(n) < 2 + 2\mu$ and $|\mu| < 1$, concluding the proof.

## B   Interpretation of Robust Momentum

To provide intuition behind robust momentum, we follow [10] and view the update rule in Eqn. 16 as a Lur'e feedback control system. For simplicity, we consider the case where, $x_t \in \mathbb{R}$ and write the update in matrix form

$$\begin{pmatrix} x_t \\ x_{t+1} \end{pmatrix} = \begin{pmatrix} 0 & 1 \\ -\mu & 1+\mu \end{pmatrix} \begin{pmatrix} x_{t-1} \\ x_t \end{pmatrix} + \begin{pmatrix} 0 \\ -\gamma \end{pmatrix} f(y_t); \quad y_t = \begin{pmatrix} -\alpha & 1+\alpha \end{pmatrix} \begin{pmatrix} x_{t-1} \\ x_t \end{pmatrix} \quad (20)$$

In the frequency domain, in order for the system 20 to be stable, we require that for all $|z| = 1$,

$$\text{Re}((1 - pz^{-1})((k-1)\tilde{G}(pz) - 1)) < 0,$$

where $\tilde{G}(pz)$ is a transformed transfer function, and $p$, a scaling factor. The imaginary axis, where $\text{Re} = 0$, is the stability boundary, and the specific construction of the parameters $\gamma$, $\mu$ and $\alpha$ pushes this boundary into the negative real axis to $-v = (1 + p)(1 - k + 2kp - kp^2)/2p$, hence achieving robust stability through the design of $\gamma$, $\mu$ and $\alpha$.

## C   Empirical Evidence

In this section, we provide the full plots presented in Section 2.3 and 6. We visualize the average norm of the SMoE and MoE outputs respectively, for 80 epochs of training in all 6 layers in Fig. 6 and 7. Notably, in all plots, less a minor exception in layer 3 of SMoE, there is a decrease in the norm of each SMoE and MoE layer output throughout training. This is consistent with our expectation that the gate learns the optimal $\alpha^*$ in a multi-objective optimization problem (Eqn. 8 and 9).

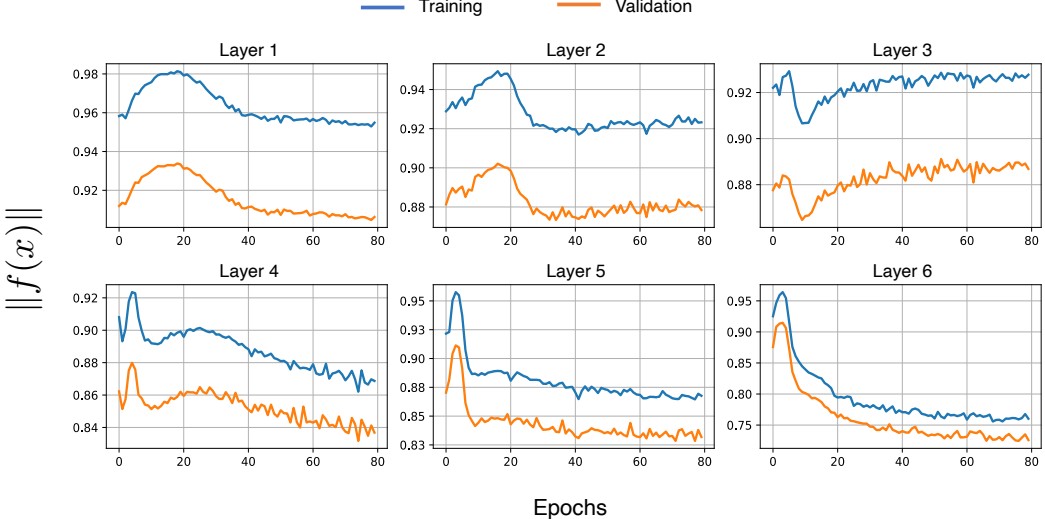

Figure 6: Average norm of the SMoE outputs at all layers during 80 epochs of training for the baseline SMoE.

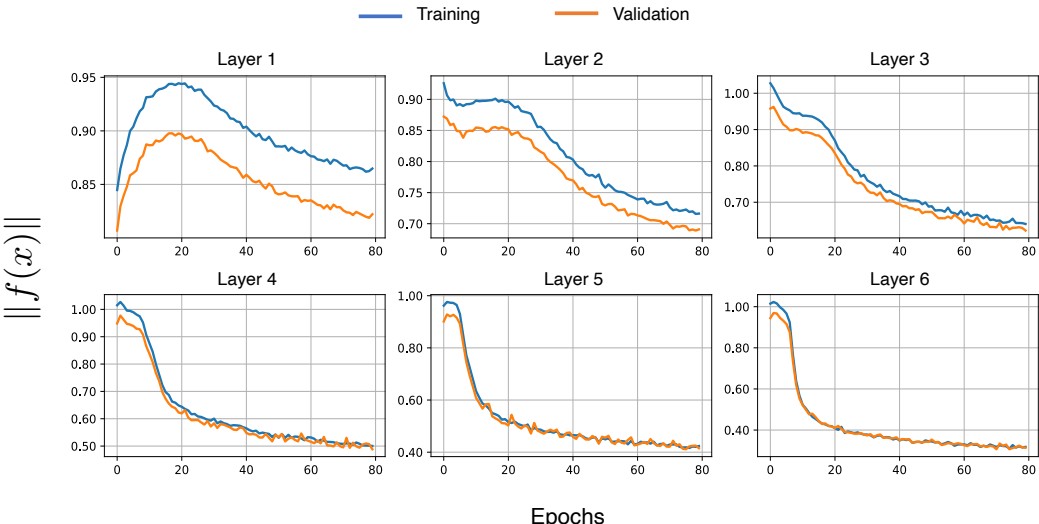

Figure 7: Average norm of the MoE outputs at all layers during 80 epochs of training for the baseline MoE.

In Fig. 8, 9 and 10, we present the proportion of time each expert is chosen in decreasing magnitude of the norm of the expert outputs for SMoE, MomentumSMoE and AdamSMoE. As discussed in Section 6, in accordance with our connection between GD and SMoE, a more even distribution of expert selection based on the size of the norm of the expert outputs should yield improved performance of the model. We demonstrate in Section 5.1 that both MomentumSMoE and AdamSMoE significantly exceed the baseline performance and correspondingly, flattens the normed-based load distribution among experts as observed in Fig. 8, 9 and 10. These strong empirical evidences serve to reinforce our optimization framework for SMoE.

## D Experiment Details

For clarity, we summarize the new models developed in this paper for each task in Table 4 and address the lacking implementations here. First, as the introduction of AdamSMoE into V-MoE leads to unstable training, we defer this challenge to future work. Second, our primary goal when studying the Momentum-Soft MoE model is to showcase the universality of our MomentumSMoE method

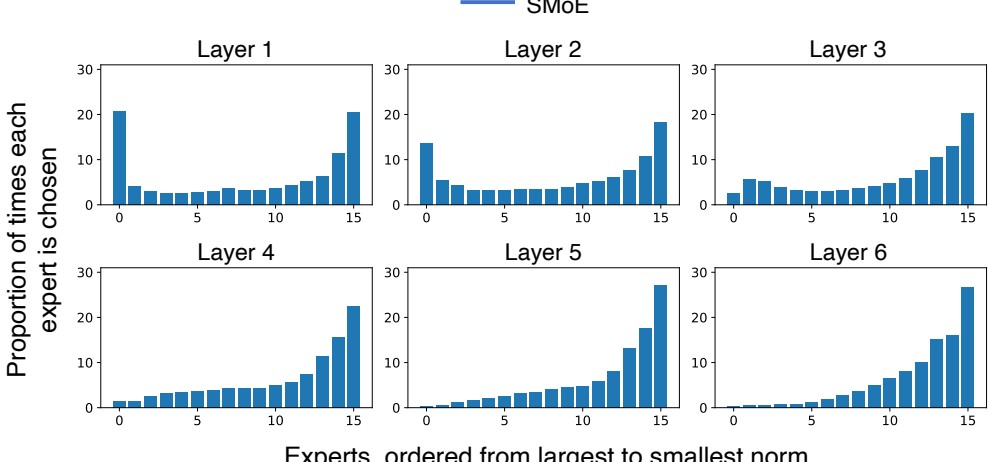

Figure 8: Proportion of each expert chosen, ordered from the largest norm of each expert output to the smallest norm, in all layers of baseline SMoE.

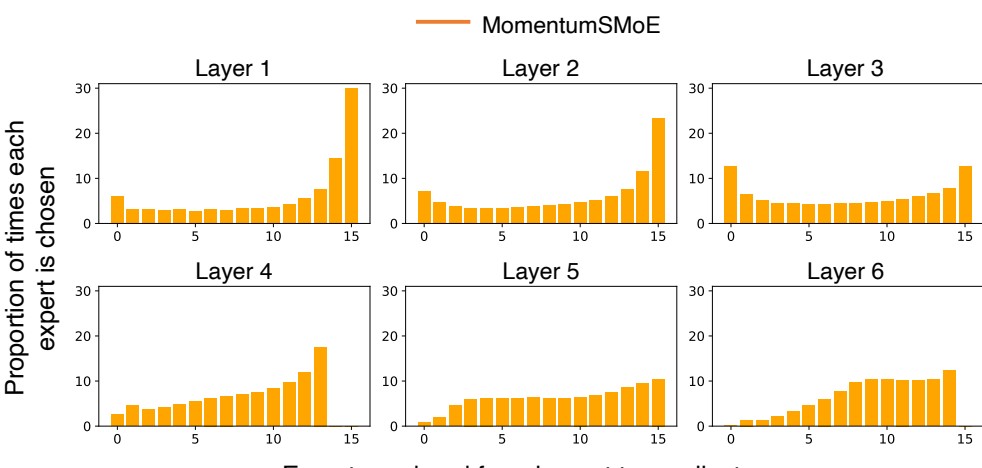

Figure 9: Proportion of each expert chosen, ordered from the largest norm of each expert output to the smallest norm, in all layers of MomentumSMoE.

Table 4: A summary of the new models developed for WikiText-103 language modeling and ImageNet-1K object recognition tasks.

| Task | Model/Method | MomentumSMoE | AdamSMoE | Robust MomentumSMoE |
|---|---|:---:|:---:|:---:|
| WikiText-103 | SMoE GLaM | ✓ | ✓ | ✓ |
| ImageNet-1K ImageNet-R ImageNet-A ImageNet-C | V-MoE | ✓ | × | ✓ |
| ImageNet-1K | Soft MoE | ✓ | × | × |

across both SMoE and MoE, hence we did not integrate AdamW and Robust Momentum into Soft MoE. We leave these implementations for future work.

### D.1 WikiText-103 Language Modeling

**Dataset:** The WikiText-103 dataset [43] is derived from Wikipedia articles and is designed to capture long-range contextual dependencies. The training set contains about 28,000 articles, with a

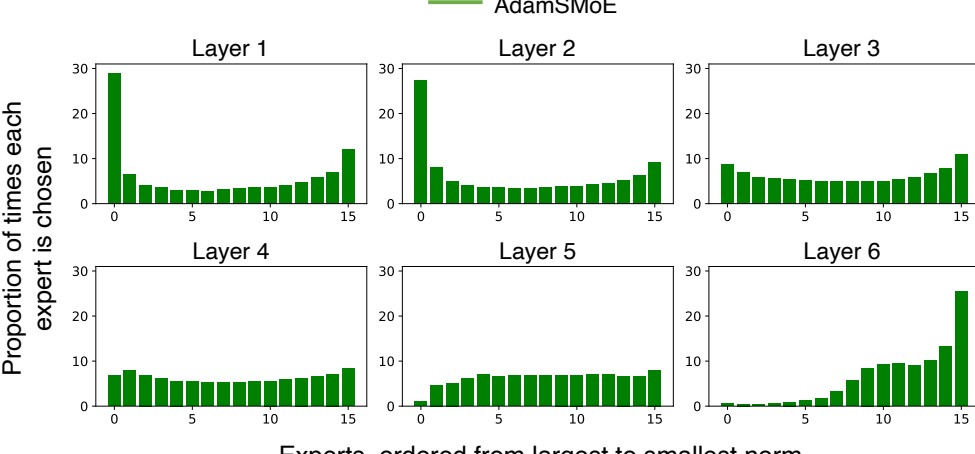

Figure 10: Proportion of each expert chosen, ordered from the largest norm of each expert output to the smallest norm, in all layers of AdamSMoE.

total of 103 million words. Each article is divided into text blocks with approximately 3,600 words. The validation and test sets have 218,000 and 246,000 words, respectively, with both sets comprising 60 articles and totaling about 268,000 words. On the attacked dataset, we corrupt the both validation and test data to demonstrate the robustness of MomentumSMoE and AdamSMoE using TextAttack's word swap attack [46]. This adversarial attack randomly replaces words in the dataset with a generic "AAA" for evaluation making it difficult for the model to predict the next word in the sequence correctly.

**Model and baselines:**   We use the Switch Transformer [18], referred to as SMoE in our tables and figures, and GLaM [16] baselines, which replaces each multilayer perceptron (MLP) layer and every other MLP layer in a vanilla language modeling transformer with a SMoE layer, respectively. For MomentumSMoE, we replace each MLP layer with a MomentumSMoE layer and initialise each momentum vector, $p_0$ at 0. We do the same for MomentumGLaM with every other MLP layer.

For consistency, we define the number of layers in each model as the number of SMoE layers. The default model used in each experiment is medium sized with 6 layers, but we include a comparison between a smaller model with 3 layers as well as a larger one with 12 layers in Appendix E.2. Each model has 16 experts in every SMoE, MomentumSMoE and AdamSMoE layer and selects 2 experts ($K = 2$) per input. All models use the same sparse router function consisting of a linear network receiving the input data followed by the TopK, then the Softmax function. The small models train for 60 epochs, the medium and large SMoE models train for 80 epochs and the GLaM models train for 120 epochs without any additional load balancing loss. Our implementation is based on the code base developed by [54], publicly available at https://github.com/ofirpress/sandwich_transformer and https://github.com/giangdip2410/CompeteSMoE/tree/main.

**AdamSMoE/AdamGLaM:**   It is observed that Adam has certain divergent behavior during large-scale training leading to unstable loss curves [8, 44]. In line with this observation, during the initial implementations of AdamSMoE, we experience similar instability. A widely used solution to this, proposed by [32], is to switch from Adam to gradient descent at a suitable point during training. We follow suit and use AdamSMoE in the first layer of the model and MomentumSMoE for the subsequent layers. We find that implementing AdamSMoE in the first layer is enough to significantly improve the model's overall performance and accelerate its convergence.

### D.2   ImageNet-1K Object Recognition

**Datasets:**   We use the ImageNet-1K dataset that contains 1.28M training images and 50K validation images. There are 1000 classes of images and the model learns an object recognition task. For robustness to common corruptions, we use ImageNet-C (IN-C) [25] which consists of 15 different types of corruptions applied to the ImageNet-1K validation set with 5 levels of severity. We provide a breakdown of our results on each corruption type and the mean Corruption Error (mCE) across

```
Hyperparameters: mu

def MomentumSMoE(x, momentum):
    momentum = - SMoE(x) + mu * momentum
    x = x + gamma * momentum
    return x
```

Figure 11: Pseudocode for MomentumSMoE implementation in python with 1 hyperparameter $\mu$.

```
Hyperparameters: mu, beta, eps = 1e-8

def AdamSMoE(x, gradient, squared_gradient):
    gradient = - (1 - mu) * SMoE(x) + mu * momentum
    squared_gradient = beta * squared_gradient + (1 - beta) * SMoE(x) ** 2
    x = x + gamma / (torch.sqrt(squared_gradient) + eps) * gradient - k * x
    return x
```

Figure 12: Pseudocode for AdamSMoE implementation in python with 2 hyperparameters $\mu$ and $\beta$.

escalating levels of severity for two corruption types in Appendix E.3. To test robustness to input data distribution shifts, we use ImageNet-A (IN-A) [26]. IN-A contains a 200 class subset of ImageNet-1K classes with adversarially filtered images. Finally, we test our model on ImageNet-R (IN-R) [24] which contains various artistic renditions of images. This evaluates the model's generalization ability to abstract visual renditions.

**Metrics:** On ImageNet-1K, ImageNet-A and ImageNet-R, we report the top-1 accuracies for all experiments. On ImageNet-C, the standard metric for evaluation is the mCE. To calculate this, we average the top-1 error rate for each corruption type across the 5 levels of severity and divide them by AlexNet's average errors, then take the final average across all corruption types. The direction of increasing or decreasing values of these metrics signifying greater robustness will be indicated in the table with an arrow.

**Model and baselines:** We use a small Vision Mixture of Experts (V-MoE) [62] model as the SMoE baseline for ImageNet-1K object recognition task as well as the standard robustness benchmarks. This variant of V-MoE consists of 8 Vision Transformer (ViT) blocks [15] with every odd block's MLP being replaced by a SMoE layer. In turn, in MomentumV-MoE, we replace every other MLP layer with a MomentumSMoE layer and similarly for Robust MomentumV-MoE. For all vision SMoE models, we select 2 experts ($K = 2$) at every SMoE layer for each patch. We follow the training configurations and setting as in [62] and their code base is available here https://github.com/google-research/vmoe/.

For our MoE baseline on clean ImageNet-1K data, we use Soft Mixture of Experts (Soft MoE) [56]. A Soft MoE model is designed to side step the challenging discrete optimization problem of assigning each token to an expert, as in SMoE, through a soft token assignment. Instead of each token being routed to one expert, in a Soft MoE layer, each expert is assigned a certain number of slots and each slot is allocated a weighted average of all tokens. These weights are dependant on both the tokens and the experts. In this case, Soft MoE is not considered as a SMoE, but instead a MoE. We use the smallest model, Soft MoE-tiny with 12 layers. The first 6 layers consists of standard ViT blocks and the last 6 layers replace the MLP in those blocks with a Soft MoE layer. In Momentum-Soft MoE, we implement the momentum parameter into each Soft-MoE layer as in a SMoE layer.

As there were no training details provided for Soft MoE on ImageNet-1K, we follow the training procedure in [69] for 120 epochs, using their published code base https://github.com/facebookresearch/deit. We train Momentum-Soft MoE and the baseline model using the PyTorch implementation of Soft MoE at https://github.com/bwconrad/soft-moe.

### D.3 Pseudocode

In this section, we provide the pseudocode as written in python for MomentumSMoE, AdamSMoE, and Robust MomentumSMoE for clarification on our implementation. These are found in Figures 11, 12 and 13 respectively.

```
Hyperparameters: p, L, m

def RobustMomentumSMoE(x, momentum):
    k = L / m
    gamma = k * ((1 - p) ** 2) * (1 + p) / L
    mu = k * p ** 3 / (k - 1)
    alpha = p ** 3 / ((k - 1) * ((1 - p) ** 2) * (1 + p))
    y = x + alpha * gamma * momentum
    momentum = - SMoE(y) + mu * momentum
    x = x + gamma * momentum
    return x
```

Figure 13: Pseudocode for Robust MomentumSMoE implementation in python with 3 hyperparameters $p$, $L$ and $m$.

Table 5: Perplexity (PPL) results on clean and attacked WikiText-103 validation and test data for standard MomentumSMoE with tuned hyperparameters $\mu$ and $\gamma$ and MomentumSMoE trained with different learning settings for $\mu$ and $\gamma$ that do not require tuning: *i)* Both $\mu$ and $\gamma$ are scalar learnable parameters in Pytorch. *ii)* Only $\gamma$ is learned with fixed $\mu = 0.7$.

| Model | $\mu$ | $\gamma$ | WikiText-103 | Valid PPL $\downarrow$ | Test PPL $\downarrow$ |
|---|---|---|---|---|---|
| *SMoE (baseline)* | - | - | Clean | 33.76 | 35.55 |
| | | | Attacked | 42.24 | 44.19 |
| MomentumSMoE | 0.7 | 1.0 | Clean | 32.29 | **33.46** |
| | | | Attacked | 40.94 | 42.33 |
| MomentumSMoE *(i)* | Learnable | Learnable | Clean | **32.28** | 33.87 |
| | | | Attacked | 40.41 | 42.32 |
| MomentumSMoE *(ii)* | 0.7 | Learnable | Clean | **32.28** | 33.69 |
| | | | Attacked | **39.96** | **41.84** |

# E  Additional Experimental Results

## E.1  Hyperparameters

In this section, we discuss two additional methods to avoid tuning MomentumSMoE hyperparameters for ease of implementation and efficiency.

**Time-varying momentum:**  Another form of the classical momentum proposed by [48] replaces the constant momentum parameter with a time-varying one $t - 1/t + 2$, which removes the need to choose an appropriate momentum hyperparameter. We adopt this modification into MomentumSMoE to replace $\mu$ while keeping $\gamma$ fixed at 1.0, and the MomentumSMoE time-varying (TV) layer is as follows

$$\boldsymbol{p}_t = -f(\boldsymbol{x}_t) + \frac{t-1}{t+2}\boldsymbol{p}_{t-1}; \quad \boldsymbol{x}_{t+1} = \boldsymbol{x}_t + \gamma\boldsymbol{p}_t$$

**Zero-order hold $\mu$ and $\gamma$:**  Recall Eqn. 10, the proposed accelerated MomentumSMoE layer

$$\boldsymbol{p}_t = -f(\boldsymbol{x}_t) + \mu\boldsymbol{p}_{t-1}; \quad \boldsymbol{x}_{t+1} = \boldsymbol{x}_t + \gamma\boldsymbol{p}_t.$$

When we replace MLP layers with MomentumSMoE layers, as we do in our language model MomentumSMoE, each expert function is a linear network. Explicitly expressing the MomentumSMoE layer results in the following expression

$$\boldsymbol{p}_t = (-\sum_{i=1}^{E} g(\boldsymbol{x}_t)_i \boldsymbol{W}_i^\top)\boldsymbol{x}_t + \mu\boldsymbol{p}_{t-1} = \bar{\boldsymbol{B}}\boldsymbol{x}_t + \mu\boldsymbol{p}_{t-1}; \quad \boldsymbol{x}_{t+1} = \boldsymbol{x}_t + \gamma\boldsymbol{p}_t$$

and can be interpreted as a state space representation with constant state and output matrices, $\mu$ and $\gamma$. From this perspective, we experiment with learning a discretized $\mu_t$ and $\gamma_t$ by applying the Zero-order hold (ZOH) such that they become adaptive parameters as is a common practise in optimization algorithms. $\mu$ and $\gamma$ are then parameterized as follows

$$\mu_t = e^{\Delta\mu}; \quad \gamma_t = (\Delta\mu)^{-1}(e^{\Delta\mu} - 1)\Delta\gamma$$

Table 6: Perplexity (PPL) results on clean and attacked WikiText-103 validation and test data for standard MomentumSMoE with tuned hyperparameters $\mu$ and $\gamma$ and MomentumSMoE trained with different learning settings for $\mu$ and $\gamma$ that do not require tuning: *i)* Both $\mu$ and $\gamma$ are scalar learnable parameters in Pytorch. *ii)* Only $\gamma$ is learned with fixed $\mu = 0.7$. *iii)* Only $\gamma$ is learned as a linear network then composed with a sigmoid function with fixed $\mu$. Included are the time-varying MomentumSMoE (TV) and Zero-order hold MomentumSMoE (ZOH).

| Model | $\mu$ | $\gamma$ | WikiText-103 | Valid PPL ↓ | Test PPL ↓ |
|---|---|---|---|---|---|
| *SMoE (baseline)* | - | - | Clean | 33.76 | 35.55 |
| | | | Attacked | 42.24 | 44.19 |
| MomentumSMoE | 0.7 | 1.0 | Clean | 32.29 | **33.46** |
| | | | Attacked | 40.94 | 42.33 |
| MomentumSMoE *(i)* | Learnable | Learnable | Clean | 32.28 | 33.87 |
| | | | Attacked | 40.41 | 42.32 |
| MomentumSMoE *(ii)* | 0.7 | Learnable | Clean | 32.28 | 33.69 |
| | | | Attacked | **39.96** | **41.84** |
| MomentumSMoE *(iii)* | 0.7 | Linear network | Clean | 32.30 | 33.96 |
| | | | Attacked | 40.35 | 42.39 |
| MomentumSMoE (TV) | $t-1/t+2$ | 1.0 | Clean | 33.02 | 35.08 |
| | | | Attacked | 41.44 | 43.97 |
| MomentumSMoE (ZOH) | $e^{\Delta\mu}$ | $(e^{\Delta\mu}-1)\Delta\gamma/\Delta\mu$ | Clean | **32.21** | 34.00 |
| | | | Attacked | 40.74 | 42.70 |

where $\Delta$ is the Softplus function of a learned scalar parameter, $\mu$ a learned scalar parameter and $\gamma$ a linear network with scalar outputs.

**Results:** We report the PPL for MomentumSMoE (TV) and MomentumSMoE (ZOH) on clean and word swap attacked WikiText-103 validation and test data in Table 6, an extended version of Table 5. We also include an additional setting of learning $\gamma$ as the sigmoid of a linear network. In this setting, every input has a different learning rate. While these models do improve over the baseline, there is no advantage over the standard MomentumSMoE and the other learnable $\mu$ and $\gamma$ settings. Hence, we do not recommend implementations in this section over those and keep the results here for reference.

### E.2   WikiText-103 Language Modeling

We present the results of all three sizes of SMoE and MomentumSMoE models in Table 7 and observe that with increasing model depth, the effectiveness of the momentum parameter in improving model performance increases as well. This aligns with the analogy of each layer being a GD step and with a higher number of iterations, the momentum term becomes more effective. While beneficial for large models, as implementing MomentumSMoE is computationally cost efficient, and hence, minimally affected by model depth, we note that there is an adverse effect in small models such as SMoE-small with only 3 layers. We hypothesize that the primary reason for this are the insufficient layers for the momentum term to make a positive impact and is a limitation of our method.

### E.3   ImageNet-C Full Results

We provide the full results of the top-1 accuracy and mCE of all 15 corruption types in ImageNet-C for V-MoE, MomentumV-MoE, Robust MomentumV-MoE and Sharpness Aware MomentumV-MoE (SAM-V-MoE), developed in Appendix E.4, in Table 8. Included in Figure 14 is a plot of the mCE with escalating severity of impulse noise and gaussian noise corruption, which illustrates the advantages of our methods with increasing data corruption. We observe that across all 15 corruption types, except for motion blur, Robust MomentumV-MoE outperforms the baseline V-MoE, with as high as a 6.5% increase in top-1 accuracy and 8 mCE decrease on fog corruption.

### E.4   Further Extensions Beyond Adam and Robust Momentum

The advantage of an optimization perspective for SMoEs are the countably many descent algorithms available that can be used to improve the model. In this section, we elaborate on six more extensions

Table 7: Perplexity (PPL) of baseline SMoE-small/medium/large, MomentumSMoE-small/medium/large, AdamSMoE-medium, baseline GLaM-medium, MomentumGLaM-medium and AdamGLaM-medium on clean and attacked WikiText-103 validation and test data.

| Model/Metric | Clean WikiText-103 | | Attacked WikiText-103 | |
|---|---|---|---|---|
| | Valid PPL ↓ | Test PPL ↓ | Valid PPL ↓ | Test PPL ↓ |
| *SMoE-small (baseline)* | **84.26** | **84.81** | **98.60** | **99.29** |
| MomentumSMoE-small | 85.71 | 86.65 | 100.26 | 101.18 |
| *SMoE-medium (baseline)* | 33.76 | 35.55 | 42.24 | 44.19 |
| MomentumSMoE-medium | 32.29 | 33.46 | 40.94 | 42.33 |
| AdamSMoE-medium | **31.59** | **33.25** | **39.27** | **41.11** |
| *GLaM-medium (baseline)* | 36.37 | 37.71 | 45.83 | 47.61 |
| MomentumGLaM-medium | 33.87 | 35.29 | 42.15 | 43.64 |
| AdamGLaM-medium | **32.99** | **34.32** | **41.09** | **42.81** |
| *SMoE-large (baseline)* | 29.31 | 30.33 | 36.77 | 37.83 |
| MomentumSMoE-large | **27.58** | **29.03** | **35.21** | **36.78** |

Table 8: Top-1 accuracy (%) and mean corruption error (mCE) of V-MoE, MomentumV-MoE, Robust MomentumV-MoE and SAM-V-MoE on each corruption type in ImageNet-C.

| Corruption Type | Model/Metric | *V-MoE (Baseline)* | MomentumV-MoE | Robust MomentumV-MoE | SAM-V-MoE |
|---|---|---|---|---|---|
| Brightness | Top-1 ↑ | 67.25 | 67.77 | **67.79** | 67.34 |
| | mCE ↓ | 58.01 | 57.08 | **57.06** | 57.85 |
| Contrast | Top-1 ↑ | 36.94 | 40.94 | **42.71** | 41.64 |
| | mCE ↓ | 73.91 | 69.22 | **67.15** | 68.41 |
| Defocus Blur | Top-1 ↑ | 39.89 | 41.01 | **41.02** | 40.26 |
| | mCE ↓ | 73.31 | 71.95 | **71.94** | 72.86 |
| Elastic Transform | Top-1 ↑ | 53.14 | 53.08 | **53.19** | 52.89 |
| | mCE ↓ | 72.53 | 72.63 | **72.45** | 72.92 |
| Fog | Top-1 ↑ | 38.23 | 42.00 | **44.85** | 44.03 |
| | mCE ↓ | 75.39 | 70.79 | **67.31** | 68.31 |
| Frost | Top-1 ↑ | 50.12 | 51.51 | **51.83** | 51.45 |
| | mCE ↓ | 60.35 | 58.66 | **58.27** | 58.73 |
| Gaussian Noise | Top-1 ↑ | 49.38 | 50.92 | **52.08** | 50.92 |
| | mCE ↓ | 57.11 | 55.37 | **54.06** | 55.36 |
| Glass Blur | Top-1 ↑ | 36.65 | 36.74 | **37.30** | 36.43 |
| | mCE ↓ | 76.67 | 76.56 | **75.88** | 76.93 |
| Impulse Noise | Top-1 ↑ | 47.72 | 49.30 | **50.59** | 49.15 |
| | mCE ↓ | 56.66 | 54.95 | **53.56** | 55.11 |
| JPEG Compression | Top-1 ↑ | 60.21 | 60.44 | **60.53** | 60.23 |
| | mCE ↓ | 65.61 | 65.23 | **65.08** | 65.58 |
| Motion Blur | Top-1 ↑ | **43.59** | 43.10 | 43.35 | 42.08 |
| | mCE ↓ | **71.77** | 72.40 | 72.08 | 73.69 |
| Pixelate | Top-1 ↑ | 61.66 | 62.61 | **63.14** | 62.30 |
| | mCE ↓ | 53.41 | 52.09 | **51.35** | 52.52 |
| Shot Noise | Top-1 ↑ | 47.96 | 49.03 | **50.51** | 49.66 |
| | mCE ↓ | 58.18 | 56.98 | **55.33** | 56.28 |
| Snow | Top-1 ↑ | 36.91 | 37.31 | **37.32** | 37.12 |
| | mCE ↓ | 72.78 | 72.32 | **72.31** | 72.54 |
| Zoom Blur | Top-1 ↑ | 35.03 | 35.88 | **36.14** | 35.20 |
| | mCE ↓ | 81.38 | 80.31 | **79.99** | 81.17 |

to MomentumSMoE, namely, Nesterov accelerated gradient [48], time-varying momentum with scheduled restart [64, 47], RMSprop [27], sharpness aware minimization [64] (SAM), negative momentum [20], and complex momentum [41]. We report their results on clean and word swap attacked WikiText-103 language modeling task in Table 9. The nature of SAM is to find local minima where the geometry of the loss landscape is flat to improve the generalization ability of the model. This aligns with the objective of improving the robustness of models to distribution shifts. Hence, we implement SAM in V-MoE as well and provide a comparison with V-MoE, MomentumV-MoE

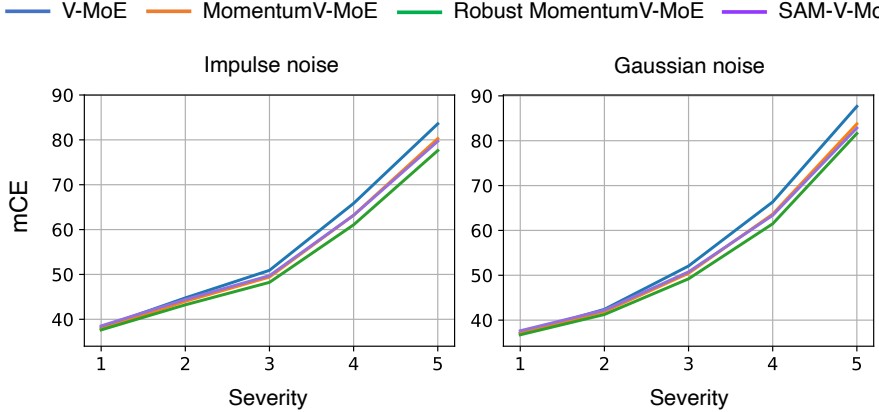

Figure 14: mCE (lower is better) of baseline V-MoE, MomentumV-MoE, Robust MomentumV-MoE and SAM-V-MoE on increasing severities of impulse noise and gaussian noise corruption. As the severity increases, the effect of momentum, robust momentum and SAM becomes increasingly apparent.

Table 9: Perplexity (PPL) results on clean and word swap attacked WikiText-103 validation and test data for baseline SMoE, NAG-SMoE, SR-SMoE, rms-SMoE, SAM-SMoE, Robust MomentumSMoE, Negative-MomentumSMoE, and Complex-MomentumSMoE.

| Model/Metric | Clean WikiText-103 | | Attacked WikiText-103 | |
|---|---|---|---|---|
| | Valid PPL ↓ | Test PPL ↓ | Valid PPL ↓ | Test PPL ↓ |
| *SMoE (baseline)* | 33.76 | 35.55 | 42.24 | 44.19 |
| NAG-SMoE | 33.83 | 35.46 | 41.94 | 43.97 |
| SR-SMoE | 32.96 | 35.01 | 41.21 | 43.72 |
| rms-SMoE | 32.43 | 34.25 | 40.58 | 42.60 |
| SAM-SMoE | 33.39 | 35.05 | 41.47 | 43.44 |
| Robust MomentumSMoE | 33.22 | 34.45 | 41.49 | 42.78 |
| Negative-MomentumSMoE | 33.48 | 35.09 | 41.68 | 43.62 |
| Complex-MomentumSMoE | **32.08** | **33.34** | **40.24** | **41.66** |

and Robust MomentumV-MoE in Table 10. In all models, we replace all SMoE layers with their corresponding newly derived layer unless stated otherwise.

**Nesterov accelerated gradient (NAG):** Nesterov accelerated gradient (NAG) [48] takes the momentum method a step further by looking ahead to where the momentum term will move the parameters and providing a correction. The foresight of the update prevents the parameters from moving in an undesirable direction too quickly, preventing large oscillations especially when the learning rate is high. We implement NAG in our SMoE gradient framework as

$$\boldsymbol{p}_t = -\gamma f(\boldsymbol{x}_t - \mu \boldsymbol{p}_{t-1}) + \mu \boldsymbol{p}_{t-1}; \quad \boldsymbol{x}_{t+1} = \boldsymbol{x}_t + \boldsymbol{p}_t$$

where $f(\boldsymbol{x}_t)$ is the SMoE output and $\mu \in (0,1)$ and $\gamma > 0$ are two hyperparameters corresponding to the momentum coefficient and step size respectively. We refer to this implementation as a NAG-SMoE.

**Time-varying momentum with scheduled restart:** An enhancement of the time-varying momentum covered in Section E.1 is to include a scheduled restart for the momentum parameter, $t - 1/t + 2$. Such a modification can help to recover an optimal convergence rate as accelerated methods do not necessarily maintain a monotonic decrease in objective value [64, 47]. As our model has 6 layers, we choose to restart after layer 3 and the RS-SMoE update is

$$\boldsymbol{p}_t = -f(\boldsymbol{x}_t) + \frac{t \mod 3}{t \mod 3 + 3} \boldsymbol{p}_{t-1}; \quad \boldsymbol{x}_{t+1} = \boldsymbol{x}_t + \gamma \boldsymbol{p}_t$$

where $f(\boldsymbol{x}_t)$ is the SMoE output and $\gamma > 0$ is the step size.

**RMSprop:** RMSprop is an adaptive step size algorithm that scales the gradient by an exponentially decaying average of the squared gradients [27]. The algorithm replaces a global learning rate, which

Table 10: Top-1 accuracy (%) and mean corruption error (mCE) of V-MoE, MomentumV-MoE, Robust MomentumV-MoE and SAM-V-MoE on validation and test ImageNet-1K data and popular standard robustness benchmarks for image classification.

| Model | Valid IN-1K Top-1 ↑ | Test IN-1K Top-1 ↑ | IN-R Top-1 ↑ | IN-A Top-1 ↑ | IN-C Top-1 ↑ | mCE ↓ |
|---|---|---|---|---|---|---|
| *V-MoE (baseline)* | 76.49 | 73.16 | 36.10 | 5.25 | 46.98 | 67.14 |
| MomentumV-MoE | **76.92** | **73.26** | 37.45 | **6.48** | 48.11 | 65.77 |
| Robust MomentumV-MoE | 76.66 | 73.20 | **37.57** | 6.37 | **48.82** | **64.92** |
| SAM-V-MoE | 76.26 | 72.84 | 36.64 | 6.27 | 48.05 | 65.88 |

is difficult to tune due to the widely differing magnitudes of the gradients of each parameter, with one that adapts to the individual parameter gradients. In addition, by keeping a running average of the squared gradients, we avoid extreme fluctuations in the step size due to stochastic batch updates. Incorporating rmsprop into the SMoE optimization perspective, we have the following update in our rms-SMoE layer,

$$\boldsymbol{p}_t = \mu \boldsymbol{p}_{t-1} + (1-\mu)f(\boldsymbol{x}_t)^2; \quad \boldsymbol{x}_{t+1} = \boldsymbol{x}_t - \frac{\gamma}{\sqrt{\boldsymbol{p}_t + \epsilon}}f(\boldsymbol{x}_t)$$

where $f(\boldsymbol{x}_t)$ is the SMoE output, $\mu \in (0,1)$ is the moving average parameter and, $\gamma > 0$ the step size. Similar to AdamSMoE, we experience some instability when using rms-SMoE to replace all SMoE layers. Hence, we follow suit, and only replace the first layer with rms-SMoE and replace subsequent SMoE layers with MomentumSMoE.

**Sharpness-Aware Minimization:** In a standard machine learning approach, training models with the usual optimization algorithms minimize the training empirical loss, which is typically non-convex. This results in multiple local minima that may have similar loss values but lead to vastly different generalization abilities. As such, models that perform well on training data, may still have inferior validation performance. From studies relating the geometry of the loss landscape to generalization, specifically, and intuitively, a flatter minima should yield greater generalization ability [17, 29, 31]. Further, this would increase the model's robustness to distribution shifts in input data or labels.

The sharpness-aware minimization (SAM) algorithm [64] aims to take advantage of this relationship and seek not just any local minimum during training, but a minima whose neighbourhood also has uniformly low loss values, in other words, lower sharpness, to improve robustness. Implementing the algorithm with the $l_2$-norm in the SMoE optimization framework yields the SAM-SMoE layer

$$\hat{\epsilon}(\boldsymbol{x}_t) = \rho f(\boldsymbol{x}_t)/\|f(\boldsymbol{x}_t)\|_2; \quad \boldsymbol{p}_t = f(\boldsymbol{x}_t + \hat{\epsilon}(\boldsymbol{x}_t)); \quad \boldsymbol{x}_{t+1} = \boldsymbol{x}_t - \gamma \boldsymbol{p}_t$$

where $f(\boldsymbol{x}_t)$ is the SMoE output, $\rho > 0$ is a hyperparameter controlling the size of the neighbourhood and, $\gamma > 0$ the step size.

**Complex and Negative Momentum:** Classic momentum works well in a gradient-based optimization when the Jacobian of our update step, as a fixed-point operator, has real eigenvalues. However, this might not always be the case. Negative momentum, which simply chooses negative momentum parameters, is preferred by operators with complex eigenvalues [20]. In situations where the spectrum is purely imaginary or has mixtures of complex and imaginary eigenvalues, complex momentum is more robust than both classic and negative momentum [41]. This is due to its oscillations at fixed frequencies between adding and subtracting the momentum term. We oscillate the sign of the momentum term by choosing a complex momentum parameter and then updating our weights by only the real part of the momentum term. This translates to the following update in the Complex-MomentumSMoE layer:

$$\boldsymbol{p}_t = -f(\boldsymbol{x}_t) + \mu \boldsymbol{p}_{t-1}; \quad \boldsymbol{x}_{t+1} = \boldsymbol{x}_t + \gamma \Re(\boldsymbol{p}_t), \tag{21}$$

where $\mu \in \mathbb{C}$ and $\gamma > 0$ are hyperparameters corresponding to the momentum coefficient and step size, respectively and $\Re$ extracts the real component of the momentum term.

**Results:** On the language modeling task, though all models, except NAG-SMoE on clean WikiText-103 validation data, do outperform the baseline across all metrics, most have only marginal gains that fall short of the performance gap achieved with MomentumSMoE and AdamSMoE. This is

Table 11: Run time per sample, memory and number of parameters of MomentumSMoE and AdamSMoE as compared to the baseline SMoE during training and test time.

| Model | Sec/Sample (Training) | Sec/Sample (Test) | Memory (Training) | Memory (Test) | Parameters |
|---|---|---|---|---|---|
| *SMoE (baseline)* | 0.0315 | 0.0303 | 22168MB | 17618MB | 216M |
| MomentumSMoE | 0.0317 | 0.0304 | 22168MB | 17618MB | 216M |
| AdamSMoE | 0.0321 | 0.0307 | 22168MB | 17618MB | 216M |

Table 12: Total computation time for SMoE, MomentumSMoE and AdamSMoE to reach 38 PPL on WikiText-103 validation data.

| Model | Time (minutes) |
|---|---|
| *SMoE (baseline)* | 85.56 |
| MomentumSMoE | **81.77** |
| AdamSMoE | 84.23 |

with the exception of Complex-MomentumSMoE, which has an even greater improvement over the baseline than MomentumSMoE. Complex-MomentumSMoE's enhanced results further verify the power and promise of our momentum-based framework in designing SMoE. On the computer vision task, we find that SAM-V-MoE does perform relatively well on corruption benchmarks, exceeding the baseline by more than 1% on ImageNet-A and ImageNet-C.

### E.5 Comparison of Computational Efficiency

A common consideration when introducing modified layers into deep models is the potential increase in computational overhead. We aim to alleviate that concern by providing the run time per sample, memory and number of parameters of MomentumSMoE and AdamSMoE as compared to the baseline SMoE during both training and test time in Table 11. We also provide the total computation time required for all models to reach the same PPL level on WikiText-103 validation data in Table 12. We observe that MomentumSMoE and AdamSMoE are comparable to the baseline SMoE across all metrics at both training and test time with negligible computational cost.

## F   Broader Impacts

Our research enhances both clean data handling and robust performance, particularly in socially impactful domains. Notably, we demonstrate improved results in object recognition, benefiting self-driving cars, and language modeling, enhancing AI chatbot assistants. We show significant advancements in resisting data perturbation, aiming to protect critical AI systems from malicious actors. Furthermore, we achieve competitive performance in language modeling with contaminated data, reflecting real-world scenarios where data is often imperfect. While the potential for AI misuse exists, our work provides substantial improvements in fundamental architectures and theory, which we hope will lead to further socially beneficial outcomes.

