# OpenReview forum: "MomentumSMoE: Integrating Momentum into Sparse Mixture of Experts"
_NeurIPS.cc/2024/Conference — NeurIPS 2024 poster_

### Official Review · Reviewer_GFpG · 2024-07-09

**Soundness:** 3
**Presentation:** 3
**Contribution:** 4
**Rating:** 7
**Confidence:** 4

**Summary:**

The paper introduces MomentumSMoE, a novel integration of heavy-ball momentum into Sparse Mixture of Experts (SMoE) to enhance stability and robustness. It establishes a connection between SMoE and gradient descent on multi-objective optimization problems.
The paper demonstrates theoretical and empirical improvements of MomentumSMoE over standard SMoE across various tasks. The method is universally applicable to many SMoE models, including V-MoE and GLaM, with minimal additional computational cost.

**Strengths:**

To the best of my knowledge, attempting to accelerate the fixed point iteration in SMoE is an original idea.

It seems like there is comprehensive empirical evidence for the method, but I am not an expert on metrics for the SMoE, and will have to rely on other reviews to be confident in this strength.

The paper is fairly clear, with well-organized sections and figures.

**Weaknesses:**

My largest negative for this paper is the largely unfounded connection between the SMoE and gradient descent.  If the authors had made a connection to accelerating fixed-point iterations in general, I would want to accept this paper.  Essentially, the authors are assuming that $\nabla_x f$ has strictly real eigenvalues when they should just work with truly, potentially complex, eigenvalues, ex., using tools as in Azizian et. al.  For example, when performing this analysis, various other acceleration schemes are often better, like negative momentum (Gidel et. al.) or complex momentum (Lorraine et. al.).  I would be curious to see some empirical investigation (or theoretical) or what the eigenvalues of $\nabla_x f$ are – ex., as in Figure 7 of https://arxiv.org/pdf/2102.08431 -- to validate any theoretical claims about what acceleration schemes should be used.

But, of course, the spectrum is only known in small-scale problems, leading to the second weakness, which is that some of the methods – ex., RobustSMoE – seem to rely on knowing the spectrum to set various parameters, which we won’t have access in real settings.  Th

The theoretical results are also largely just reproductions of known theoretical results for momentum once you assume that the update from the SMoE is a gradient. This makes them not much of a contribution from my point of view other than leveraging existing tools. I think these results could be easily substituted for analogous techniques from Azizian.

Azizian, Waïss, et al. "Accelerating smooth games by manipulating spectral shapes." International Conference on Artificial Intelligence and Statistics. PMLR, 2020.
Lorraine, Jonathan P., et al. "Complex momentum for optimization in games." International Conference on Artificial Intelligence and Statistics. PMLR, 2022.
Gidel, Gauthier, et al. "Negative momentum for improved game dynamics." The 22nd International Conference on Artificial Intelligence and Statistics. PMLR, 2019.

**Questions:**

How can the assumptions about the fixed point operator's spectrum and the Jacobian's conservativeness be validated or relaxed in practical scenarios?

Are there more general acceleration tools than momentum you might want to use for this problem?

**Limitations:**

The limitations are discussed briefly, but a delineated section elaborating on all the limitations would be valuable.

---

> ### Author Rebuttal · Authors · 2024-08-06
>
> **Q1. Unfounded connection between the SMoE and gradient descent. Connection to accelerating fixed-point iterations. The authors should work with complex eigenvalues [Azizian et. al.] using negative (Gidel et. al.) or complex momentum (Lorraine et. al.). Empirical (or theoretical) investigation and an analysis of the eigenvalues of $\nabla_x f$  is needed.**
>
> **Answer:** We respectfully disagree with the reviewer’s comment that the connection between SMoE and gradient descent (GD) is largely unfounded. In Section 2.3, from line 128 to line 144, we discuss the empirical evidences (Fig. 2, 3) for connection between SMoE and GD.
>
> GD updates can be considered as fixed-point iteration: at optimality where the gradient is 0, GD finds a fixed-point solution. Thus, techniques in accelerating fixed-point iterations can be used to find the equilibrium point of SMoE. Following the reviewer's suggestion, we have conducted additional experiments that incorporate negative momentum (Gidel, 2019) and complex momentum (Lorraine, 2022) into SMoE using our momentum-based design framework. Table 2 in the attached PDF compares the PPL of NegativeMomentumSMoE and ComplexMomentumSMoE with MomentumSMoE, AdamSMoE, and SMoE on the clean/attacked WikiText-103. Our AdamSMoE achieves the best PPL. ComplexMomentumSMoE obtains slightly better PPL than MomentumSMoE (positive momentum coefficient), while both of these methods significantly outperform NegativeMomentumSMoE and the SMoE baseline.
>
> Furthermore, when analyzing the eigenvalues of $\nabla_x f$, where $f$ is the output of an SMoE layer in the trained SMoE above, we observe that the mean absolute real and imaginary parts of the eigenvalues are 4.06 and 0.34, averaged over all SMoE layers in the model, respectively. (Gidel, 2019) and (Lorraine, 2022) suggest that positive momentum converges when the eigenvalues are real, while negative momentum has better convergence when the eigenvalues have large imaginary parts. Also, the convergence of the complex momentum method is robust to the wider range of complex eigenvalues, including purely real and imaginary ones. Since the imaginary part of the eigenvalues of $\nabla_x f$ is small compared to their real part (0.34 vs. 4.06), NegativeMomentumSMoE should not work well, and MomentumSMoE with positive momentum should be a better design choice. Since ComplexMomentumSMoE can handle a wider range of complex eigenvalues, it helps improve the performance of MomentumSMoE. However, due to the small imaginary part of eigenvalues of $\nabla_x f$, the improvement is small. These explain our results in Table 2 in the attached PDF and we also provide a plot of the spectrum in Figure 1.
>
> We would like to thank the reviewer for your suggestion on negative and complex momentum. The significant improvement of MomemtunSMoE, AdamSMoE, and ComplexMomentumSMoE over the SMoE baseline further verifies the power and promise of the momentum-based framework for designing SMoE. Moreover, given our framework, analyzing the eigenvalues of $\nabla_x f$ offers a principled way to select the right momentum method for SMoE design.
>
> **Q2. Methods like RobustSMoE seem to rely on knowing the spectrum to set various parameters, which we won’t have access in real settings.**
>
> **Validating/Relaxing assumptions about the fixed point operator's spectrum and the Jacobian's conservativeness**
>
> **Answer:**  The motivation behind the design of complex momentum is to work over a large range of eigenvalues, as it is common that, in practice, we do not know the spectrum [Lorraine, 2022]. Similarly, for MomentumSMoE and AdamSMoE, we need to tune the hyperparameters, i.e., $\mu$, $\gamma$, as well as $\beta$ in AdamSMoE, in order to achieve convergence.
>
> In contrast, the Robust Momentum Method in [Cyrus, 2018] that we use to develop Robust MomentumSMoE **does not require the spectrum information**, and the convergence of the algorithm **does not depend on the spectrum values**. This Robust Momentum Method can be considered as a Lur’e feedback control system, and the hyperparameters $\gamma$, $\mu$, and $\alpha$ (see Eqn. (18), our manuscript) are designed to push the stability boundary into the negative real axis. This allows an additional margin for the stability conditions of the system to hold.
>
> **Q3. The novelty of theoretical results?**
>
> **Answer:** As mentioned in line 156 in our manuscript, we are inspired by the techniques in [Qian, 1999] and adapt these techniques to prove the convergence and stability results of Momentum SMoE (Proposition 1 and Corollary 1). The techniques we use are not new, and other techniques, such as those in [Azizian, 2020] or [Wilson, 2018], can be leveraged to prove similar results. Our objective is to provide theoretical guarantees on the convergence and stability of MomentumSMoE to prove that MomentumSMoE is more stable than the SMoE baseline, which justifies the empirical advantages of MomentumSMoE over SMoE shown in our manuscript.
>
> **Q4. More general acceleration tools than momentum?**
>
> **Answer:** Momentum methods, such as Nesterov’s accelerated gradient method, are simple approximations of the proximal point method [Ahn, 2022]. For future work, we can employ the proximal point method to develop better SMoE models. In our manuscript, we also explore the sharpness-aware minimization (SAM) [Roulet, 2017] approach to enhance the robustness of SMoE (lines 805 - 821 in Appendix E.4).
>
> **References**
>
> O’donoghue, B, et al. Adaptive restart for accelerated gradient schemes. 2015.
>
> Lucas, J., et al. Aggregated momentum. 2018.
>
> Cyrus, S., et al. A robust accelerated optimization algorithm for strongly convex functions. 2018
>
> Qian, N. On the momentum term in gradient descent learning algorithms. 1999.
>
> Wilson, A. C., et al. A Lyapunov analysis of accelerated methods in optimization. 2021.
>
> Ahn, K., et al.Understanding nesterov's acceleration via proximal point method. 2022.
>
> Roulet, V., et al. Sharpness, restart and acceleration. 2017.

---

> > ### Comment · Reviewer_GFpG · 2024-08-07
> > **Response to Rebuttal**
> >
> > I appreciate the author's response and results, which ameliorate my main concerns.  As such, I am revising the rating to be higher.  Here are some elaborations on the response:
> >
> > Figure 1 in the response really strengthens the paper, and it should be referenced in the Section “Empirical Evidence for the Gradient Descent Analogy of (S)MoE.”  However, I think it's quite important that the writing clearly emphasizes that the MoE is not doing gradient descent on a single objective function since you observed complex eigenvalues.  Instead, the MoE is “close to doing gradient descent on an objective function”, in a spectral sense.  Further, all subsequent theories should be noted as a simplified heuristic justification for your method, as the theory requires assuming a conservative, real-valued vector field.  I believe the theory could be generalized to work with complex eigenvalues, and I do not think it is required from this work to generalize the theory. Still, the gap between theory and practice should be easily documented for readers.
> >
> > I believe the comparison with other fixed-point acceleration schemes also enhances the paper.
> >
> > Also, regarding robust momentum, introducing an extra hyperparameter -- even randomly -- often leads to some improvement from tuning the new parameter.  What we need to do to show this is useful is some exploration showing that doing your search in this new 3-parameter space is better than optimizing the 2-parameter space – ex., finding better values when doing a grid search of N queries in 3 dimensions instead of 2.  Or, you could compare to other ways to introduce a third hyperparameter, and show you are better.  However, I don’t think this is non-essential for publication, as the robust momentum is not the key contribution.  More, you note that various momentum flavors can be projected to this setting.
> >
> > It seems infeasible for NeurIPS, but it would improve the paper to rephrase the momentum theory to work with spectrums that are bound to be “near” the real axis in some sense. I think it's likely you could reuse theory from papers on fixed-point acceleration.
> >
> > For Figure 1, shouldn’t the spectrum be symmetric, because if I have an eigenvalue at “x + iy” then I must have an eigenvalue at “x - iy” because the eigenvalues come in conjugate pairs.  Since your plot is not symmetric, what is being shown? Are you only plotting the positive roots?
> >
> > Also, this does not affect my review, but I think is an interesting avenue for future exploration.  For your Figure 1 with the spectrum exploration, it is interesting that the spectrum is almost all positive and closely clustered to the real axis. Why is there one eigenvalue with a negative real part?  A better understanding of the shape of spectrums you encounter allows you to select optimal acceleration schemes.  Does this spectral shape hold for various MoE models?  Does the spectral shape change during training?
> > Also, there is a single eigenspace with a negative real part—this seems notable. Does this eigenspace correspond to anything interpretable? In what eigenspaces do the parameters accumulate as the iterations progress?  Answering these may lead to insights, allowing you to design better fixed-point acceleration schemes.

---

> ### Author Response · Authors · 2024-08-08
> **Thanks for your endorsement!**
>
> We would like to thank the reviewer for your further feedback, and we appreciate your endorsement.
>
> We will include Figure 1 in the Section "Empirical Evidences for the Gradient Descent Analogy of (S)MoE" in our revision. Following your suggestion, we will revise our manuscript to verify that MoE is “close to doing gradient descent on an objective function in a spectral sense” and update our stability analysis of MomentumSMoE (Section 3 in our manuscript) accordingly. We will also discuss the extension from our momentum-based framework for SMoE to accelerating fixed-point iterations and include the experiments with the ComplexMomentumSMoE and NegativeMomentumSMoE in our revision.
>
> Furthermore, we agree with the reviewer that introducing a hyperparameter into the model usually improves its performance and that a thorough comparison will help further validate our experimental results. As you mentioned, robust momentum is not our main contribution, so we leave this for future work. However, we would like to point out that when tuning the hyperparameter in RobustMomentumSMoE, we are tuning the model on clean ImageNet-1K data, where there is a slight improvement. While on perturbed datasets, such as ImageNet-A/R/C, there is a significant increase in accuracy. These results lead us to believe that the model's enhanced performance is not entirely due to the introduction of an extra parameter, but rather the design of robust momentum itself.
>
> In addition, further investigation of acceleration methods that consider different spectrums is an interesting direction that we will continue to explore. We believe that our momentum-based framework can be extended to these methods, as well as their theories, and will benefit from them.
>
> For Figure 1 in our 1-page attached PDF, the spectrum is not symmetric due to the presence of purely real eigenvalues. We also take the average over the layers in the SMoE, causing the spectrum to skew upwards in our plot.
>
> Once again, we appreciate your suggestions and feedback which help enhance the quality of our paper and introduce exciting new directions for further development of our MomentumSMoE.

---

> > ### Comment · Reviewer_GFpG · 2024-08-08
> >
> > "For Figure 1 in our 1-page attached PDF, the spectrum is not symmetric due to the presence of purely real eigenvalues. We also take the average over the layers in the SMoE, causing the spectrum to skew upwards in our plot."
> >
> > In your plot, there are complex eigenvalues -- i.e., eigenvalues with a non-zero imaginary part -- which do not have their conjugate plotted.  Why is this occurring?  It would not be caused by real eigenvalues, as every complex eigenvalue will still have its conjugate.  I also don't understand what you mean by taking the average over layers.  You are computing the eigenvalues of a real nxn matrix, which always satisfies the property of having complex eigenvalues occur in conjugate pairs.

---

> > > ### Author Response · Authors · 2024-08-08
> > > **Clarification on Figure 1 in Our Attached PDF**
> > >
> > > We are sorry for this misunderstanding. Please let us clarify Figure 1 here. By taking the average over the layers, we mean that in a medium SMoE model, there are 6 layers and 16 experts. Each of these will correspond to 1 real nxn matrix. We find the eigenvalues of each of these matrices, which are either real numbers or conjugate pairs. We find the modulus and phase of each eigenvalue and average over the 6 layers and 16 experts. Positive real eigenvalues in the spectrum result in phases of $0$ while negative real eigenvalues in the spectrum result in phases of $\pi$. Consequently, averaging these $0$ and $\pi$ phases with the phases of complex eigenvalues causes the plot to be skewed upwards.
> > >
> > > We have tried our best to reproduce Figure 7 in [Lorraine, 2022] in the context of SMoE. We would appreciate your suggestions to better make this figure.
> > >
> > > **Reference**
> > >
> > > Lorraine, Jonathan P., David Acuna, Paul Vicol, and David Duvenaud. "Complex momentum for optimization in games." International Conference on Artificial Intelligence and Statistics. PMLR, 2022.

---

> ### Comment · Reviewer_GFpG · 2024-08-08
>
> The goal is to visualize the eigenvalues of $\nabla_x f$, and perhaps view how it changes as "optimization progresses".  I think this corresponds to taking the eigenvalues of the matrix, which is an average of the matrices for each expert, and the "optimization step" is like the layer.  So the "averaging" would be done for the matrices for the experts themselves and not the eigenvalues if I understand correctly.  Further, plotting the eigenvalues' union would be a more typical visualization strategy than averaging if there are multiple distinct matrices, as getting rid of the conjugate structure is confusing to me. I think a union over the different layers may make sense, or as separate plots.
>
> You might also find it easier to visualize as a heatmap than a scatterplot. If you wanted to see a more granular structure, you could play with different ways to plot it—for example, a different figure/color for each layer/expert. For example, in [Lorraine, 2022] Figure 9 shows the spectrum at the start and the end of training.

---

> > ### Author Response · Authors · 2024-08-08
> > **Reply to Reviewer GFpG**
> >
> > Thanks for your prompt response. We appreciate your detailed and valuable suggestions to better visualize the eigenvalues of $\nabla_{x}f$. If you allow us to include an anonymous link that includes additional figures in our reply, we will be very happy to make the figures you suggested and present them in this discussion. We will also include those figures in our revision.

---

> ### Author Response · Authors · 2024-08-09
>
> On the point of averaging the matrices of each expert, we agree that this would be a better way to visualize the eigenvalues of $\nabla_{x}f$ than averaging the eigenvalues. However, as only 2 experts are chosen at each layer for each data point (i.e., our model implements top-2 SMoE), and further, a convex combination of their matrices are applied where the coefficients of this convex combination are the corresponding affinity (gate) scores, there are many possible combinations. How would you like to best visualize them? Would it make sense to you to take the 2 experts chosen the most frequently during training/inference time and plot the spectrum of their convex combination?
>
> For example, at layer $i$, $i=1,...,6$, we have 16 expert matrices $A_{i_1}, A_{i_2}, ..., A_{i_{16}}$, and the top 2 most frequently chosen matrices are $A_{i_j}$ and $A_{i_k}$. Then, for $\lambda \in [0,1]$ decided by the affinity (gate) scores, we plan to compute the convex combination $A_i = \lambda A_{i_j} + (1-\lambda) A_{i_k}$ and plot the spectrum of $A_i$ in each layer.
>
> We would be willing to provide you with the eigenvalues and a snippet of python code that can be easily run in a jupyter notebook for visualization.

---

> ### Comment · Reviewer_GFpG · 2024-08-09
>
> The goal is to visualize the eigenvalues of $\nabla_x f$ (or, similarly, the eigenvalues of the Jacobian of the fixed point operator) at each optimization step, so you should visualize exactly that.  Note that you are abusing notation with $\nabla_x f$ as it seems to assume a potential function, so, really, you are looking at the eigenvalues of the Jacobian of the expert's aggregated update.  This is uniquely defined, and there should not be ambiguity in how you establish these eigenvalues.
>
> As such, you should use the exact weighting of the expert's matrices, such that it equals $\nabla_x f$ (at some specific optimization step).  I believe this is weighting them according to the gating scores. Still, the key point is figuring out how to visualize exactly the spectrum of $\nabla_x f$.  You need to derive exactly what form the Jacobian $\nabla_x f$ -- which will involve gating values and expert matrices -- then visualize exactly that, and this will tell you exactly how to weigh the different matrices uniquely.  You should not just choose arbitrary averages of the matrices or arbitrary averages of eigenvalues of different terms.
>
> Note that the eigenvalues will likely be different at each step of your fixed-point operator. It is more common to visualize the union of them or them separately at varying points in optimization or only at the end of optimization, which dictates asymptotic convergence rates.
>
> This specific spectrum is useful, as it dictates the spectrum of the Jacobian of the fixed point operator, which bounds on convergence rates to the fixed point and, relatedly, the optimal acceleration schemes (like momentum and its variants).  See Theorem 1 from Negative momentum (https://arxiv.org/pdf/1807.04740), which just re-states the classic result of Prop 4.4.1 from Bertsekas 1999.
>
> D. P. Bertsekas. Nonlinear programming. Athena
> scientific Belmont, 1999.

---

> > ### Author Response · Authors · 2024-08-10
> >
> > Thanks for your detailed explanation and informative reply. We follow the reviewer's suggestion and derive the Jacobian $\nabla_x f$ as follows.
> >
> > In Eqn. (9) in our manuscript, the SMoE is formulated as
> >
> > $$
> > x_{t+1} = x_t - \gamma\sum_{i=1}^K \text{softmax}(\text{TopK}(g_{i}(x_t))) [-u_i(x_t)] = x_t - \gamma f(x_t),
> > $$
> >
> > where, again, $f(x) = \nabla_{x}F = \sum_{i=1}^K \text{softmax}(\text{TopK}(g_{i}(x))) [-u_i(x)]$ plays the role of a fixed point operator and $F$ is the objective function defined in Eqn. (3) in our manuscript. The Jacobian $\nabla_x f$ is then given by:
> >
> > $$
> > \nabla_x f = \sum_{i=1}^K \nabla_x  \, \{\text{softmax}(\text{TopK}(g_{i}(x))) [-u_i(x)]\}.
> > $$
> >
> > For each data point, we differentiate $f$ using automatic differentiation in PyTorch. We then compute the eigenvalues of the Jacobian at layers 1, 3 and 6 in the SMoE model, corresponding to their respective steps in our optimization algorithm. These eigenvalues are listed in the code snippet below. You can run our code to visualize those eigenvalues in a scatterplot. As can be seen from the resultant figures, the eigenvalues cluster around 0, $\pi$ and $-\pi$, indicating that while they are not exactly real, the eigenvalues are close to real. It is also interesting that in the last step (layer 6), there seem to be slightly less eigenvalues with almost purely imaginary parts.

---

> > > ### Author Response · Authors · 2024-08-10
> > > **Code Snippet Part 1/4**
> > >
> > > ~~~ #python
> > > import torch
> > > import matplotlib.pyplot as plt
> > >
> > > # layer 1
> > > eig0_base = torch.tensor([ -28.21+0.00j, -18.35+0.21j, -18.35-0.21j, -14.52+0.68j, -14.52-0.68j, -7.76+1.04j, -7.76-1.04j, -15.45+0.00j, -14.99+0.12j, -14.99-0.12j, -14.32+0.00j,
> > > -12.24+0.64j, -12.24-0.64j, -4.99+1.07j, -4.99-1.07j, -12.49+0.36j, -12.49-0.36j, -9.83+0.76j, -9.83-0.76j, 1.91+1.01j, 1.91-1.01j, -10.79+0.61j,
> > > -10.79-0.61j, -7.73+0.85j, -7.73-0.85j, -11.27+0.42j, -11.27-0.42j, -11.91+0.10j, -11.91-0.10j, -10.23+0.53j, -10.23-0.53j, -6.54+0.85j, -6.54-0.85j,
> > > -4.04+0.96j, -4.04-0.96j, 7.57+0.00j, 6.55+0.47j, 6.55-0.47j, 0.15+0.98j, 0.15-0.98j, -7.79+0.71j, -7.79-0.71j, 4.91+0.65j, 4.91-0.65j,
> > > -9.64+0.14j, -9.64-0.14j, -3.78+0.85j, -3.78-0.85j, 2.30+0.79j, 2.30-0.79j, -1.28+0.88j, -1.28-0.88j, 0.89+0.82j, 0.89-0.82j, -10.34+0.16j,
> > > -10.34-0.16j, -7.92+0.60j, -7.92-0.60j, -6.47+0.67j, -6.47-0.67j, -10.16+0.00j, -7.87+0.54j, -7.87-0.54j, -8.77+0.29j, -8.77-0.29j, -2.87+0.80j,
> > > -2.87-0.80j, -0.06+0.79j, -0.06-0.79j, 3.06+0.70j, 3.06-0.70j, 5.00+0.55j, 5.00-0.55j, 5.16+0.40j, 5.16-0.40j, 6.78+0.13j, 6.78-0.13j,
> > > 6.83+0.00j, 5.82+0.31j, 5.82-0.31j, 6.09+0.19j, 6.09-0.19j, -8.94+0.00j, 2.90+0.67j, 2.90-0.67j, -4.00+0.74j, -4.00-0.74j, -2.03+0.74j,
> > > -2.03-0.74j, 0.21+0.73j, 0.21-0.73j, -5.03+0.65j, -5.03-0.65j, -8.39+0.00j, 1.27+0.66j, 1.27-0.66j, 2.54+0.62j, 2.54-0.62j, 4.75+0.43j,
> > > 4.75-0.43j, 3.28+0.54j, 3.28-0.54j, 5.78+0.16j, 5.78-0.16j, 5.42+0.24j, 5.42-0.24j, -0.75+0.69j, -0.75-0.69j, -4.10+0.61j, -4.10-0.61j,
> > > -6.44+0.47j, -6.44-0.47j, -7.08+0.35j, -7.08-0.35j, -7.75+0.18j, -7.75-0.18j, -7.46+0.27j, -7.46-0.27j, 5.36+0.00j, 4.62+0.25j, 4.62-0.25j,
> > > -2.09+0.61j, -2.09-0.61j, -5.08+0.48j, -5.08-0.48j, -5.27+0.45j, -5.27-0.45j, 4.99+0.00j, -3.61+0.55j, -3.61-0.55j, -6.72+0.24j, -6.72-0.24j,
> > > -6.60+0.22j, -6.60-0.22j, -7.18+0.01j, -7.18-0.01j, -7.04+0.06j, -7.04-0.06j, -1.82+0.59j, -1.82-0.59j, -0.18+0.59j, -0.18-0.59j, 4.10+0.28j,
> > > 4.10-0.28j, 3.61+0.35j, 3.61-0.35j, 1.72+0.50j, 1.72-0.50j, 4.05+0.13j, 4.05-0.13j, 2.38+0.45j, 2.38-0.45j, 2.71+0.39j, 2.71-0.39j,
> > > -3.14+0.51j, -3.14-0.51j, -0.91+0.56j, -0.91-0.56j, -1.11+0.54j, -1.11-0.54j, 0.36+0.51j, 0.36-0.51j, -2.56+0.51j, -2.56-0.51j, -6.10+0.00j,
> > > -4.94+0.29j, -4.94-0.29j, -4.73+0.30j, -4.73-0.30j, 2.97+0.33j, 2.97-0.33j, 4.08+0.07j, 4.08-0.07j, 3.95+0.13j, 3.95-0.13j, 0.57+0.47j,
> > > 0.57-0.47j, 1.69+0.41j, 1.69-0.41j, 3.44+0.21j, 3.44-0.21j, 2.96+0.20j, 2.96-0.20j, -0.15+0.45j, -0.15-0.45j, -0.59+0.46j, -0.59-0.46j,
> > > -2.92+0.42j, -2.92-0.42j, -5.39+0.23j, -5.39-0.23j, -4.43+0.33j, -4.43-0.33j, -3.61+0.35j, -3.61-0.35j, -5.59+0.09j, -5.59-0.09j, -3.97+0.29j,
> > > -3.97-0.29j, -4.96+0.17j, -4.96-0.17j, -5.13+0.12j, -5.13-0.12j, -5.21+0.03j, -5.21-0.03j, 0.88+0.39j, 0.88-0.39j, 2.95+0.14j, 2.95-0.14j,
> > > 2.02+0.29j, 2.02-0.29j, -2.25+0.38j, -2.25-0.38j, -1.58+0.39j, -1.58-0.39j, -2.39+0.33j, -2.39-0.33j, -0.05+0.38j, -0.05-0.38j, 0.32+0.37j,
> > > 0.32-0.37j, 1.51+0.31j, 1.51-0.31j, 1.26+0.29j, 1.26-0.29j, 3.21+0.01j, 3.21-0.01j, 2.26+0.19j, 2.26-0.19j, 2.75+0.06j, 2.75-0.06j,
> > > -3.49+0.21j, -3.49-0.21j, -3.91+0.14j, -3.91-0.14j, -4.12+0.00j, -4.04+0.04j, -4.04-0.04j, -1.50+0.33j, -1.50-0.33j, -0.80+0.35j, -0.80-0.35j,
> > > 2.50+0.09j, 2.50-0.09j, 1.70+0.21j, 1.70-0.21j, 0.99+0.26j, 0.99-0.26j, -3.46+0.13j, -3.46-0.13j, 0.17+0.26j, 0.17-0.26j, -0.02+0.27j,
> > > -0.02-0.27j, -2.45+0.25j, -2.45-0.25j, -0.99+0.27j, -0.99-0.27j, -1.47+0.28j, -1.47-0.28j, -2.07+0.23j, -2.07-0.23j, -2.83+0.16j, -2.83-0.16j,
> > > -3.11+0.11j, -3.11-0.11j, 1.85+0.15j, 1.85-0.15j, 2.20+0.00j, 2.16+0.03j, 2.16-0.03j, -3.17+0.00j, -2.25+0.18j, -2.25-0.18j, 1.36+0.16j,
> > > 1.36-0.16j, 1.79+0.07j, 1.79-0.07j, -2.97+0.00j, -2.66+0.06j, -2.66-0.06j, -1.69+0.19j, -1.69-0.19j, -0.91+0.23j, -0.91-0.23j, -1.31+0.20j,
> > > -1.31-0.20j, -2.42+0.03j, -2.42-0.03j, -2.05+0.12j, -2.05-0.12j, -0.37+0.22j, -0.37-0.22j, 0.68+0.20j, 0.68-0.20j, -0.68+0.20j, -0.68-0.20j,
> > > 1.81+0.00j, 0.05+0.19j, 0.05-0.19j, 0.55+0.16j, 0.55-0.16j, 1.37+0.00j, 1.20+0.08j, 1.20-0.08j, 0.90+0.12j, 0.90-0.12j, -1.09+0.15j,
> > > -1.09-0.15j, -1.81+0.10j, -1.81-0.10j, -1.81+0.05j, -1.81-0.05j, -0.28+0.16j, -0.28-0.16j, 1.07+0.04j, 1.07-0.04j, 0.99+0.05j, 0.99-0.05j,
> > > 0.80+0.09j, 0.80-0.09j, -1.22+0.10j, -1.22-0.10j, -0.53+0.13j, -0.53-0.13j, -1.53+0.05j, -1.53-0.05j, -1.45+0.03j, -1.45-0.03j, -0.62+0.11j,
> > > -0.62-0.11j, -0.11+0.11j, -0.11-0.11j, 0.16+0.09j, 0.16-0.09j, 0.52+0.03j, 0.52-0.03j, 0.04+0.07j, 0.04-0.07j, 0.52+0.00j, 0.22+0.04j,
> > > 0.22-0.04j, -0.93+0.05j, -0.93-0.05j, -0.42+0.05j, -0.42-0.05j, 0.19+0.00j, -0.02+0.00j, -1.12+0.00j, -0.94+0.00j, -0.65+0.00j, -0.68+0.00j])
> > > ~~~

---

> > > > ### Author Response · Authors · 2024-08-10
> > > > **Code Snippet Part 2/4**
> > > >
> > > > ~~~#python
> > > > # layer 3
> > > > eig2_base = torch.tensor([-46.60+0.00j, -17.27+0.00j, -11.27+1.00j, -11.27-1.00j, 15.41+0.00j, 14.28+0.31j, 14.28-0.31j, -13.24+0.34j, -13.24-0.34j, -13.74+0.00j, -12.13+0.48j, -12.13-0.48j, -9.66+0.79j,
> > > > -9.66-0.79j, -6.73+1.06j, -6.73-1.06j, 6.03+1.16j, 6.03-1.16j, -11.59+0.00j, 11.38+0.67j, 11.38-0.67j, -8.53+0.72j, -8.53-0.72j, -2.86+1.17j, -2.86-1.17j, 10.74+0.54j,
> > > > 10.74-0.54j, 9.84+0.64j, 9.84-0.64j, 8.81+0.74j, 8.81-0.74j, -3.41+1.02j, -3.41-1.02j, -2.00+1.04j, -2.00-1.04j, 4.80+0.95j, 4.80-0.95j, 0.95+1.05j, 0.95-1.05j,
> > > > 2.01+1.03j, 2.01-1.03j, 10.79+0.33j, 10.79-0.33j, 10.51+0.02j, 10.51-0.02j, 9.58+0.31j, 9.58-0.31j, 8.68+0.53j, 8.68-0.53j, 6.59+0.78j, 6.59-0.78j, 0.32+0.98j,
> > > > 0.32-0.98j, -5.63+0.88j, -5.63-0.88j, -10.85+0.33j, -10.85-0.33j, -9.14+0.52j, -9.14-0.52j, -9.88+0.38j, -9.88-0.38j, -10.63+0.07j, -10.63-0.07j, -4.31+0.86j, -4.31-0.86j,
> > > > 8.75+0.22j, 8.75-0.22j, 6.86+0.57j, 6.86-0.57j, -6.09+0.76j, -6.09-0.76j, -8.47+0.52j, -8.47-0.52j, -6.74+0.64j, -6.74-0.64j, 4.72+0.76j, 4.72-0.76j, 3.99+0.78j,
> > > > 3.99-0.78j, 7.07+0.46j, 7.07-0.46j, 8.62+0.04j, 8.62-0.04j, -8.81+0.31j, -8.81-0.31j, -9.04+0.13j, -9.04-0.13j, 0.41+0.84j, 0.41-0.84j, -0.70+0.84j, -0.70-0.84j,
> > > > -2.68+0.77j, -2.68-0.77j, -6.54+0.54j, -6.54-0.54j, -7.23+0.43j, -7.23-0.43j, -2.63+0.71j, -2.63-0.71j, -4.57+0.55j, -4.57-0.55j, 7.40+0.25j, 7.40-0.25j, 2.30+0.71j,
> > > > 2.30-0.71j, 0.30+0.72j, 0.30-0.72j, 4.17+0.61j, 4.17-0.61j, 5.69+0.48j, 5.69-0.48j, 6.00+0.38j, 6.00-0.38j, 4.52+0.52j, 4.52-0.52j, 7.35+0.00j, -1.39+0.70j,
> > > > -1.39-0.70j, 6.76+0.12j, 6.76-0.12j, 6.03+0.23j, 6.03-0.23j, 2.19+0.63j, 2.19-0.63j, 4.58+0.46j, 4.58-0.46j, 3.79+0.52j, 3.79-0.52j, -2.84+0.61j, -2.84-0.61j,
> > > > -7.68+0.00j, -7.45+0.15j, -7.45-0.15j, -6.98+0.27j, -6.98-0.27j, -3.79+0.54j, -3.79-0.54j, -7.11+0.05j, -7.11-0.05j, -5.99+0.35j, -5.99-0.35j, -6.31+0.28j, -6.31-0.28j,
> > > > -4.61+0.47j, -4.61-0.47j, 5.44+0.20j, 5.44-0.20j, -1.53+0.58j, -1.53-0.58j, 1.56+0.58j, 1.56-0.58j, 0.67+0.59j, 0.67-0.59j, -0.20+0.59j, -0.20-0.59j, -6.01+0.00j,
> > > > -5.20+0.29j, -5.20-0.29j, -4.55+0.36j, -4.55-0.36j, -2.16+0.54j, -2.16-0.54j, 3.15+0.47j, 3.15-0.47j, 5.63+0.06j, 5.63-0.06j, 5.05+0.26j, 5.05-0.26j, 4.02+0.35j,
> > > > 4.02-0.35j, -5.68+0.07j, -5.68-0.07j, -3.48+0.41j, -3.48-0.41j, -1.57+0.50j, -1.57-0.50j, -5.02+0.23j, -5.02-0.23j, 4.92+0.00j, 2.04+0.49j, 2.04-0.49j, 1.56+0.48j,
> > > > 1.56-0.48j, 3.29+0.36j, 3.29-0.36j, 4.04+0.24j, 4.04-0.24j, -4.53+0.25j, -4.53-0.25j, -2.23+0.40j, -2.23-0.40j, -4.83+0.02j, -4.83-0.02j, 4.18+0.18j, 4.18-0.18j,
> > > > 4.11+0.14j, 4.11-0.14j, 2.30+0.36j, 2.30-0.36j, 0.68+0.43j, 0.68-0.43j, -0.38+0.43j, -0.38-0.43j, -0.70+0.43j, -0.70-0.43j, -2.98+0.35j, -2.98-0.35j, -3.33+0.33j,
> > > > -3.33-0.33j, -4.04+0.22j, -4.04-0.22j, -4.40+0.08j, -4.40-0.08j, 3.82+0.08j, 3.82-0.08j, 2.92+0.25j, 2.92-0.25j, 0.94+0.38j, 0.94-0.38j, -0.12+0.36j, -0.12-0.36j,
> > > > -1.38+0.36j, -1.38-0.36j, -2.21+0.33j, -2.21-0.33j, -3.19+0.26j, -3.19-0.26j, -4.05+0.00j, -3.84+0.13j, -3.84-0.13j, -3.50+0.16j, -3.50-0.16j, -1.58+0.34j, -1.58-0.34j,
> > > > 2.05+0.30j, 2.05-0.30j, 3.73+0.00j, 3.29+0.10j, 3.29-0.10j, -3.47+0.08j, -3.47-0.08j, -2.78+0.20j, -2.78-0.20j, -0.64+0.33j, -0.64-0.33j, -0.21+0.34j, -0.21-0.34j,
> > > > 0.40+0.31j, 0.40-0.31j, 1.79+0.27j, 1.79-0.27j, 2.17+0.22j, 2.17-0.22j, 2.48+0.17j, 2.48-0.17j, 2.75+0.04j, 2.75-0.04j, 2.42+0.14j, 2.42-0.14j, 0.67+0.29j,
> > > > 0.67-0.29j, 1.02+0.26j, 1.02-0.26j, -2.32+0.19j, -2.32-0.19j, -1.39+0.24j, -1.39-0.24j, -0.33+0.27j, -0.33-0.27j, -2.95+0.07j, -2.95-0.07j, -2.45+0.13j, -2.45-0.13j,
> > > > -2.82+0.03j, -2.82-0.03j, -2.79+0.00j, -0.57+0.24j, -0.57-0.24j, -0.90+0.21j, -0.90-0.21j, -1.10+0.18j, -1.10-0.18j, -1.72+0.13j, -1.72-0.13j, -1.79+0.10j, -1.79-0.10j,
> > > > 0.88+0.21j, 0.88-0.21j, 0.13+0.21j, 0.13-0.21j, 2.16+0.10j, 2.16-0.10j, 2.35+0.00j, 2.17+0.03j, 2.17-0.03j, 1.93+0.10j, 1.93-0.10j, 1.64+0.14j, 1.64-0.14j,
> > > > 0.71+0.17j, 0.71-0.17j, 1.39+0.11j, 1.39-0.11j, 1.53+0.07j, 1.53-0.07j, 1.76+0.01j, 1.76-0.01j, -2.01+0.00j, -1.83+0.04j, -1.83-0.04j, -1.44+0.08j, -1.44-0.08j,
> > > > -0.07+0.17j, -0.07-0.17j, -0.75+0.14j, -0.75-0.14j, -0.51+0.16j, -0.51-0.16j, 0.72+0.12j, 0.72-0.12j, -1.58+0.00j, -1.06+0.10j, -1.06-0.10j, -1.35+0.00j, -1.32+0.03j,
> > > > -1.32-0.03j, -0.13+0.13j, -0.13-0.13j, 0.92+0.06j, 0.92-0.06j, 1.18+0.02j, 1.18-0.02j, 0.34+0.11j, 0.34-0.11j, 0.84+0.01j, 0.84-0.01j, -0.62+0.08j, -0.62-0.08j,
> > > > 0.16+0.08j, 0.16-0.08j, -0.23+0.09j, -0.23-0.09j, 0.62+0.03j, 0.62-0.03j, 0.20+0.05j, 0.20-0.05j, -0.90+0.00j, -0.53+0.02j, -0.53-0.02j, -0.26+0.05j, -0.26-0.05j, -0.24+0.00j])
> > > > ~~~

---

> > > > > ### Author Response · Authors · 2024-08-10
> > > > > **Code Snippet Part 3/4**
> > > > >
> > > > > ~~~#python
> > > > > # layer 6
> > > > > eig5_base = torch.tensor([ 2.64+1.62j, 2.64-1.62j, -16.31+0.77j, -16.31-0.77j, -17.69+0.09j, -17.69-0.09j, -16.41+0.37j, -16.41-0.37j, -7.03+1.32j, -7.03-1.32j, -12.89+0.83j,
> > > > > -12.89-0.83j, 11.58+0.26j, 11.58-0.26j, 11.76+0.04j, 11.76-0.04j, 9.81+0.68j, 9.81-0.68j, 1.99+1.23j, 1.99-1.23j, 0.71+1.20j, 0.71-1.20j,
> > > > > -4.73+1.19j, -4.73-1.19j, -1.76+1.14j, -1.76-1.14j, -8.65+1.06j, -8.65-1.06j, -6.52+1.10j, -6.52-1.10j, -3.00+1.12j, -3.00-1.12j, -12.92+0.64j,
> > > > > -12.92-0.64j, -14.16+0.48j, -14.16-0.48j, -10.68+0.80j, -10.68-0.80j, -15.07+0.00j, -7.74+0.91j, -7.74-0.91j, -11.19+0.62j, -11.19-0.62j, -13.21+0.32j,
> > > > > -13.21-0.32j, -13.05+0.00j, -12.84+0.08j, -12.84-0.08j, -8.55+0.79j, -8.55-0.79j, -11.98+0.28j, -11.98-0.28j, 2.43+1.05j, 2.43-1.05j, 0.87+1.02j,
> > > > > 0.87-1.02j, 5.79+0.84j, 5.79-0.84j, 7.31+0.63j, 7.31-0.63j, 3.47+0.94j, 3.47-0.94j, 2.38+0.95j, 2.38-0.95j, 5.07+0.81j, 5.07-0.81j,
> > > > > 8.30+0.45j, 8.30-0.45j, 9.42+0.00j, 8.86+0.13j, 8.86-0.13j, 7.75+0.41j, 7.75-0.41j, -5.44+0.93j, -5.44-0.93j, -2.96+0.92j, -2.96-0.92j,
> > > > > -10.01+0.47j, -10.01-0.47j, -10.96+0.00j, -10.29+0.25j, -10.29-0.25j, -5.31+0.73j, -5.31-0.73j, -6.92+0.61j, -6.92-0.61j, 0.95+0.88j, 0.95-0.88j,
> > > > > 5.88+0.58j, 5.88-0.58j, 4.95+0.67j, 4.95-0.67j, 7.76+0.07j, 7.76-0.07j, 3.51+0.73j, 3.51-0.73j, -9.62+0.00j, -9.44+0.00j, 6.15+0.31j,
> > > > > 6.15-0.31j, 0.37+0.84j, 0.37-0.84j, 1.72+0.74j, 1.72-0.74j, -2.77+0.74j, -2.77-0.74j, -4.83+0.73j, -4.83-0.73j, -8.99+0.00j, -7.44+0.44j,
> > > > > -7.44-0.44j, -5.92+0.59j, -5.92-0.59j, 7.24+0.00j, 6.34+0.25j, 6.34-0.25j, -2.43+0.74j, -2.43-0.74j, -7.86+0.28j, -7.86-0.28j, -8.53+0.08j,
> > > > > -8.53-0.08j, -7.30+0.29j, -7.30-0.29j, 5.34+0.41j, 5.34-0.41j, 6.51+0.16j, 6.51-0.16j, 6.51+0.00j, -4.92+0.49j, -4.92-0.49j, -2.04+0.63j,
> > > > > -2.04-0.63j, 0.72+0.63j, 0.72-0.63j, 3.77+0.51j, 3.77-0.51j, 2.83+0.54j, 2.83-0.54j, 5.87+0.00j, 5.03+0.23j, 5.03-0.23j, 4.88+0.10j,
> > > > > 4.88-0.10j, 3.61+0.38j, 3.61-0.38j, -6.84+0.04j, -6.84-0.04j, -6.94+0.00j, -5.80+0.31j, -5.80-0.31j, -6.03+0.19j, -6.03-0.19j, 0.66+0.57j,
> > > > > 0.66-0.57j, 0.01+0.58j, 0.01-0.58j, -3.12+0.51j, -3.12-0.51j, -2.70+0.53j, -2.70-0.53j, -0.77+0.55j, -0.77-0.55j, -4.14+0.40j, -4.14-0.40j,
> > > > > 4.56+0.09j, 4.56-0.09j, 2.87+0.40j, 2.87-0.40j, 1.61+0.50j, 1.61-0.50j, -5.38+0.12j, -5.38-0.12j, -4.35+0.34j, -4.35-0.34j, -3.38+0.41j,
> > > > > -3.38-0.41j, -1.32+0.49j, -1.32-0.49j, 0.34+0.47j, 0.34-0.47j, -4.58+0.25j, -4.58-0.25j, -5.38+0.00j, 3.55+0.26j, 3.55-0.26j, 3.23+0.29j,
> > > > > 3.23-0.29j, -4.87+0.00j, -1.79+0.46j, -1.79-0.46j, 1.35+0.42j, 1.35-0.42j, 0.38+0.45j, 0.38-0.45j, 3.69+0.08j, 3.69-0.08j, 3.33+0.17j,
> > > > > 3.33-0.17j, 0.72+0.42j, 0.72-0.42j, -2.44+0.39j, -2.44-0.39j, -3.58+0.27j, -3.58-0.27j, -4.32+0.18j, -4.32-0.18j, -4.83+0.09j, -4.83-0.09j,
> > > > > -4.77+0.00j, -2.68+0.33j, -2.68-0.33j, -3.98+0.08j, -3.98-0.08j, -1.17+0.36j, -1.17-0.36j, 2.79+0.20j, 2.79-0.20j, 0.92+0.33j, 0.92-0.33j,
> > > > > 3.30+0.00j, 3.13+0.04j, 3.13-0.04j, -0.09+0.35j, -0.09-0.35j, 1.96+0.25j, 1.96-0.25j, 2.61+0.14j, 2.61-0.14j, -1.34+0.31j, -1.34-0.31j,
> > > > > -2.04+0.28j, -2.04-0.28j, -2.48+0.25j, -2.48-0.25j, -3.01+0.15j, -3.01-0.15j, -3.68+0.00j, -0.08+0.29j, -0.08-0.29j, 1.42+0.23j, 1.42-0.23j,
> > > > > 1.89+0.18j, 1.89-0.18j, -3.32+0.00j, -3.03+0.07j, -3.03-0.07j, -2.68+0.00j, -1.82+0.23j, -1.82-0.23j, -1.23+0.26j, -1.23-0.26j, -2.22+0.17j,
> > > > > -2.22-0.17j, -2.34+0.14j, -2.34-0.14j, 1.01+0.24j, 1.01-0.24j, 2.39+0.03j, 2.39-0.03j, -0.34+0.25j, -0.34-0.25j, -1.49+0.18j, -1.49-0.18j,
> > > > > 1.89+0.12j, 1.89-0.12j, 1.97+0.02j, 1.97-0.02j, 1.40+0.16j, 1.40-0.16j, 1.00+0.20j, 1.00-0.20j, 0.01+0.21j, 0.01-0.21j, 0.67+0.21j,
> > > > > 0.67-0.21j, -0.83+0.18j, -0.83-0.18j, -1.95+0.10j, -1.95-0.10j, -2.07+0.03j, -2.07-0.03j, 1.13+0.12j, 1.13-0.12j, 1.35+0.08j, 1.35-0.08j,
> > > > > 0.87+0.11j, 0.87-0.11j, -0.51+0.15j, -0.51-0.15j, -2.04+0.00j, -1.30+0.11j, -1.30-0.11j, -1.57+0.00j, -1.23+0.08j, -1.23-0.08j, -0.24+0.14j,
> > > > > -0.24-0.14j, 0.31+0.14j, 0.31-0.14j, 1.19+0.05j, 1.19-0.05j, 1.30+0.00j, 1.27+0.00j, -0.65+0.12j, -0.65-0.12j, -1.41+0.02j, -1.41-0.02j,
> > > > > -1.33+0.00j, 0.50+0.10j, 0.50-0.10j, 0.03+0.11j, 0.03-0.11j, -0.95+0.06j, -0.95-0.06j, 0.79+0.06j, 0.79-0.06j, 0.88+0.04j, 0.88-0.04j,
> > > > > -0.63+0.06j, -0.63-0.06j, -0.45+0.09j, -0.45-0.09j, -0.92+0.02j, -0.92-0.02j, -0.83+0.00j, -0.12+0.07j, -0.12-0.07j, 0.63+0.02j, 0.63-0.02j,
> > > > > 0.35+0.05j, 0.35-0.05j, -0.69+0.03j, -0.69-0.03j, -0.27+0.06j, -0.27-0.06j, -0.44+0.05j, -0.44-0.05j, 0.45+0.02j, 0.45-0.02j, 0.40+0.02j,
> > > > > 0.40-0.02j, 0.03+0.04j, 0.03-0.04j, -0.47+0.00j, -0.01+0.02j, -0.01-0.02j, -0.09+0.02j, -0.09-0.02j, 0.03+0.01j, 0.03-0.01j, -0.32+0.00j])
> > > > > ~~~

---

> > > > > > ### Author Response · Authors · 2024-08-10
> > > > > > **Code Snippet Part 4/4**
> > > > > >
> > > > > > ~~~#python
> > > > > > # layer 1
> > > > > > abs0 = torch.log(eig0_base.abs())
> > > > > > phase0 = eig0_base.angle()
> > > > > >
> > > > > > # layer 3
> > > > > > abs2 = torch.log(eig2_base.abs())
> > > > > > phase2 = eig2_base.angle()
> > > > > >
> > > > > > # layer 6
> > > > > > abs5 = torch.log(eig5_base.abs())
> > > > > > phase5 = eig5_base.angle()
> > > > > >
> > > > > > fig, ax = plt.subplots(1, 3, figsize=(16,4), sharey=True)
> > > > > > fontsize = 15
> > > > > > linewidth = 4
> > > > > > ax[0].scatter(abs0, phase0, s = 30, c = "blue", alpha=0.7,edgecolors="blue")
> > > > > > ax[1].scatter(abs2, phase2, s = 30, c = "orange", alpha=0.7,edgecolors="orange")
> > > > > > ax[2].scatter(abs5, phase5, s = 30, c = "green", alpha=0.7,edgecolors="green")
> > > > > > ax[0].set_ylim([-3.2,3.2])
> > > > > > ax[0].set_yticks(ticks=[-3.14, -1.57, 0, 1.57, 3.14])
> > > > > > ax[0].set_yticklabels(labels=[-3.14, -1.57, 0, 1.57, 3.14], fontsize=fontsize-2)
> > > > > > ax[0].grid()
> > > > > > ax[1].grid()
> > > > > > ax[2].grid()
> > > > > > ax[0].set_title("Layer 1", fontsize=fontsize)
> > > > > > ax[0].tick_params(axis='both', which='major', labelsize=fontsize-2)
> > > > > > ax[1].set_title("Layer 3", fontsize=fontsize)
> > > > > > ax[1].tick_params(axis='both', which='major', labelsize=fontsize-2)
> > > > > > ax[2].set_title("Layer 6", fontsize=fontsize)
> > > > > > ax[2].tick_params(axis='both', which='major', labelsize=fontsize-2)
> > > > > > plt.tight_layout()
> > > > > > plt.show()
> > > > > > ~~~

---

> ### Comment · Reviewer_GFpG · 2024-08-14
>
> I have run the code you provided, and those visualizations look good.  Perhaps a heatmap would look reasonable in the plots, but that is a personal preference. This is, of course, not required for this paper, but for future analysis, might I suggest looking at the eigenvalues of fixed point operator (as in Figure 4, https://arxiv.org/pdf/1807.04740), which combined information about your optimization scheme, with the spectrum of $\nabla_x f$.  For example, for gradient descent which uses an update of $x = x - f$, for a fixed point operator $g(x) = x - f$, then the you would want to look at $eigs(\nabla_x g) = eigs(I - \nabla_x f)$.  The eigenvalues of the fixed point operator directly tell you convergence rates (in idealized scenarios), and looking at how your optimizer warps the spectrum may provide insight into the best acceleration schemes. You might want to look at the union of the eigenvalues over multiple different "layers" and optimization setups / different problems for the experts.  It could also be worth looking at what eigenspaces your updates are actually accumulating in, as this is non-convex (so most theoretical results will say it doesn't converge).  Overall, I am happy with your response, and I think this a fun topic to ponder.
>
> ```fig, ax = plt.subplots(1, 3, figsize=(16, 4), sharey=True)
> fontsize = 15
> bins = 51  # Number of bins for the heatmap. Better visualization if this number is odd imo
>
> # Heatmap for the first set of data
> h0 = ax[0].hist2d(abs0, phase0, bins=bins, cmap='Blues')
> ax[0].set_ylim([-3.2, 3.2])
> ax[0].set_yticks(ticks=[-3.14, -1.57, 0, 1.57, 3.14])
> ax[0].set_yticklabels(labels=[-3.14, -1.57, 0, 1.57, 3.14], fontsize=fontsize-2)
> ax[0].grid()
> ax[0].set_title("Layer 1", fontsize=fontsize)
> ax[0].tick_params(axis='both', which='major', labelsize=fontsize-2)
> fig.colorbar(h0[3], ax=ax[0])
>
> # Heatmap for the second set of data
> h2 = ax[1].hist2d(abs2, phase2, bins=bins, cmap='Oranges')
> ax[1].grid()
> ax[1].set_title("Layer 3", fontsize=fontsize)
> ax[1].tick_params(axis='both', which='major', labelsize=fontsize-2)
> fig.colorbar(h2[3], ax=ax[1])
>
> # Heatmap for the third set of data
> h5 = ax[2].hist2d(abs5, phase5, bins=bins, cmap='Greens')
> ax[2].grid()
> ax[2].set_title("Layer 6", fontsize=fontsize)
> ax[2].tick_params(axis='both', which='major', labelsize=fontsize-2)
> fig.colorbar(h5[3], ax=ax[2])
>
> plt.tight_layout()
> plt.show()

---

> > ### Author Response · Authors · 2024-08-14
> > **Thanks for your endorsement!**
> >
> > Thanks the reviewer for your constructive responses and interesting suggestions on the spectrum analysis of the fixed point operator. We are grateful for the time taken to provide good insights into further improvements to our paper. We will continue to explore these new perspectives in our future work. We appreciate your endorsement and the fruitful discussions.

---

### Official Review · Reviewer_puL2 · 2024-07-11

**Soundness:** 3
**Presentation:** 3
**Contribution:** 3
**Rating:** 7
**Confidence:** 3

**Summary:**

This paper proposes a variant of sparse mixture of experts, MomentumSMoE, by incorporating momentum into the traditional sparse mixture of experts framework. The authors provide both theoretical proofs and empirical evidence demonstrating that MomentumSMoE offers greater stability and robustness compared to the standard sparse mixture of experts. Experiments on language modeling and object recognition tasks are conducted to verify the effectiveness of the proposal.

**Strengths:**

1. The idea of integrating momentum into sparse mixture of experts  is interesting.
2. Both the theoretical proof and extensive empirical results are provided to demonstrate that the proposed MomentumSMoE is more stable and robust than SmoE; the experimental results are appealing.
3. The code is provided.

**Weaknesses:**

The pseudocode may be provided to better illustrate the implementation of the proposal.

**Questions:**

1. Why the MomentumV-MoE and Robust MomentumV-MoE have marginal gains on clean IN-1K data, is there any in-depth analysis available on this?
2. In the ImageNet-1K Object Recognition experiment, why was the popular top-5 accuracy metric not used, as it was in the Soft Mixture of Experts experiment?
3. As stated in the weaknesses, the authors could provide pseudocode to better clarify their proposal.

**Limitations:**

The authors adequately addressed the limitations.

---

> ### Author Rebuttal · Authors · 2024-08-06
>
> **Q1. Why the MomentumV-MoE and Robust MomentumV-MoE have marginal gains on clean IN-1K data, is there any in-depth analysis available on this?**
>
> **Answer:** V-MoE's result reported in Table 2 in our manuscript is among the state-of-the-art results on clean IN-1k data for the models that have around 45M effective parameters without pretraining. To further improve this result is challenging, and the almost 0.43\% improvement of our MomentumV-MoE vs. the V-MoE baseline is already nontrivial. Notice that both our MomentumV-MoE and Robust MomentumV-MoE significantly improve over the V-MoE baseline on robustness benchmarks, such as IN-R, IN-A, and IN-C, as shown in Table 2.
>
>
> **Q2. In the ImageNet-1K Object Recognition experiment, why was the popular top-5 accuracy metric not used, as it was in the Soft Mixture of Experts experiment?**
>
> **Answer:** Thanks for your comment. We provide the top-5 accuracy for ImageNet-1K object recognition task in Table 1 in the attached PDF.
>
> **Q3. As stated in the weaknesses, the authors could provide pseudocode to better clarify their proposal.**
>
> **Answer:** Thanks for your suggestion. We include the pseudocode for our MomentumSMoE, AdamSMoE, and Robust MomentumSMoE below.
> ~~~
> Hyperparameters: mu
>
> def MomentumSMoE(x, momentum):
> 	momentum = - SMoE(x) + mu * momentum
> 	x = x + gamma * momentum
> 	return x
> ~~~
>
> ~~~
> Hyperparameters: mu, beta, eps = 1e-8
>
> def AdamSMoE(x, gradient, squared_gradient):
> 	gradient = - (1 - mu) * SMoE(x) + mu * momentum
> 	squared_gradient = beta * squared_gradient + (1 - beta) * SMoE(x) ** 2
> 	x = x + gamma / (torch.sqrt(squared_gradient) + eps) * gradient - k * x
> 	return x
> ~~~
>
> ~~~
> Hyperparameters: p, L, m
>
> def RobustMomentumSMoE(x, momentum):
> 	k = L / m
> 	gamma = k * ((1 - p) ** 2) * (1 + p) / L
> 	mu = k * p ** 3 / (k - 1)
> 	alpha = p ** 3 / ((k - 1) * ((1 - p) ** 2) * (1 + p))
> 	y = x + alpha * gamma * momentum
> 	momentum = - SMoE(y) + mu * momentum
> 	x = x + gamma * momentum
> 	return x
> ~~~

---

> ### Author Response · Authors · 2024-08-10
> **Any Questions from Reviewer puL2 on Our Rebuttal?**
>
> We would like to thank the reviewer again for your thoughtful reviews and valuable feedback.
>
> We would appreciate it if you could let us know if our responses have addressed your concerns and whether you still have any other questions about our rebuttal.
>
> We would be happy to do any follow-up discussion or address any additional comments.

---

### Official Review · Reviewer_oJG3 · 2024-07-11

**Soundness:** 2
**Presentation:** 3
**Contribution:** 3
**Rating:** 5
**Confidence:** 4

**Summary:**

The paper introduces a novel approach to enhancing the robustness and stability of Sparse Mixture of Experts (SMoE) models. Inspired by the analogy of gradient descent and SMoE, the authors develop a family of models by incorporating momentum into the training process. The key idea is that training SMoE is a multi-objective optimization problem where the monument-based gradient descent method is more stable and robust than the vanilla one. They proposed the AdamSMoE and Robust MomentumSMoE, which demonstrate improved performance across a variety of tasks, including language modeling and object recognition.

**Strengths:**

(1) The integration of momentum into SMoE is a non-trivial innovation that addresses instability and inefficiency issues in existing models.

(2) The paper provides convincing empirical evidence showing the effectiveness of MomentumSMoE across multiple benchmarks.

(3) The proposed method's compatibility with other momentum-based optimizers, like Adam, suggests it can be broadly applied to various SMoE architectures.

**Weaknesses:**

(1) Formulating SMoE as a multi-objective optimization problem is doubtful to me. Every expert network is continually changing during the model training, which makes each objective nonstatic, which violates the basic assumption of multi-objective optimization, whose objectives should be very clear and stable.

(2) It is unconvincing to use ||f(x)|| as the key metrics to measure the efficacy of SMoE or MoE. This confuses me a lot. Please explain why the output norm represents the goodness/badness of the model.

(3) There are some grammar issues. Please use `` instead of " in the paper (line 665).

(4) There is no sufficient discussion of computation overhead. Training efficiency is a critical issue for current foundation model training. Does computation significantly increase by applying momentum over the SMoE? Keeping an additional copy weight (p in Fig 1) would take additional memory and may decrease the throughput.

I'd like to hear a more insightful discussion regarding all the points above from the authors.

**Questions:**

(1) Please explain more of line 140 ("Thus, it is expected that these two terms learn to reduce ...).

---

> ### Author Rebuttal · Authors · 2024-08-06
>
> **Q1. Formulating SMoE as a multi-objective optimization problem is doubtful to me.**
>
> **Answer:** We believe there is a misunderstanding of our formulation of SMoE as a multi-objective optimization problem. Please allow us to clear this misunderstanding by clarifying the role of expert networks in our multi-objective optimization (MOO) framework for Sparse Mixture of Experts (SMoE). We rewrite the minimization problem in Eqn. (3) as follows:
>
> $$
> \min_{x\in D, \theta \in \Theta} F(x, \theta) := \sum_{i=1}^E c_i F_i(x, \theta^{i}).
> $$
> where $\theta^{i}$ is the parameters of the expert network $i$, $\theta = \{\theta^{1},\dots,\theta^{E}\}$, and $\Theta$ is the parameter space. Also, $D$ is the feasible region, and $c_i \in \mathbb{R}$, $i=1,\dots,E$, are weights representing the importance of each objective function. In order to solve this optimization problem, we employ the alternating minimization approach as follows:
>
> *Step 1 - fix $\theta$ and minimize $F$ w.r.t. $x$*:
>
> $$
> x_{t+1} = x_{t} - \gamma \sum_{i=1}^{E}\alpha_{i}^*\nabla_xF_i(x_t,\theta^{i}_{t})
> $$
>
> $$
> = x_{t} - \gamma \sum_{i=1}^{E}\alpha_{i}^* f_{i}(x_{t}, \theta_{t}^i),
> $$
>
> where $\alpha^*=(\alpha_1^*,\dots,\alpha_E^*)$ satisfy the Pareto-stationary condition in Definition 1 in our manuscript, $\gamma$ is the step size, and $f_{i}=\nabla_xF_i$.
>
> *Step 2 - fix $x$ and minimize $F$ with respect to $\theta$*
> $$
> \theta_{t+1}^{i} = \min_{\theta^{i} \in \Theta^{i}}  c_i F_i(x_{t+1}, \theta^{i}), i=1,\dots,E.
> $$
> where $\theta^{i}$ lies in the parameter space $\Theta^{i}$.
>
> In our formulation of SMoE as a multi-objective optimization problem, we regard $-u_i(x_t)$, where $u_i$ is the $i^{\text{th}}$ expert network,  as the gradient $\nabla_{x}F_i(x_t,\theta^{i}_{t})$ in Step 1, and the score $\text{softmax}(g_i(x_t))$ from the router $g(x_t)$ is learned to approximate $\alpha_i^*$. Step 1 in the alternating minimization algorithm above corresponds to forward pass of an SMoE, and step 2 corresponds to an update step of the model's parameters (i.e., similar to an M-step in an EM algorithm). Note that this parameter update step is implicitly performed at each training iteration via optimizing the objective loss at the final layer of the model using backpropagation and gradient descent. Results in Fig 2. and the discussion from line 137 to line 144 in our manuscript provide supporting evidence for this implicit parameter update.
>
> In Step 1, the gradient $\nabla_{x}F_i(x_t,\theta_{t}^{i})$ is a function of $\theta_{t}^{i}$ and is supposed to change during the model training, which matches the observation that expert network is continually changing during the model training. Also, the objective function of our MOO problem is $F(x, \theta)$ in which we want to find $x^*$ and $\theta^*$ to minimize $F$. This objective function is static, clear, and stable.
>
> **Q2. Why $||f(x)||$ as the key metrics to measure the efficacy of (S)MoE?**
>
> **Please explain more of line 140.**
>
> **Answer:** In Fig. 2 and 3 in our manuscript, $||f(x)||$ is not used as a key metric to measure the efficacy of SMoE/MoE. Instead, the results in Fig. 2 and 3 are to justify the connection between SMoE/MoE and gradient descent. In particular, from Eqn. (8) and (9), the output of (S)MoE corresponds to $-f(x_t)$, where $f(x_t) = \nabla_xF_(x_t)$ and is the gradient of the objective function $F$ with respect to $x$ at $x_t$. If (S)MoE indeed performs (stochastic) gradient descent in its forward pass, then the norm of the (S)MoE output must decrease when t (i.e., layer index) increases since the gradient norm $\||f(x_t)\||$ decreases when gradient descent updates are applied. Fig. 3 confirms this by showing that the norm of the (S)MoE output decreases over layers in a 6-layer (S)MoE model trained on the WikiText-103 language modeling task. At the last layer, the norm increases might be due to overshooting, a common phenomenon that can occur when using gradient descent.
>
> Additionally, from line 116 to line 119 in our manuscript, we hypothesize that the scores $\text{softmax}(g_{i}(x_t))$ for MoE and $\text{softmax}(\text{TopK}(g_{i}(x_t)))$ for SMoE from the router $g(x_t)$ are learned to approximate $\alpha_i^*$. If this is indeed true, then in order to satisfy the Pareto-stationary condition in Definition 1 in our manuscript, the scores should be learned to minimize the norm $\|\sum_{i=1}^E \alpha_i \tilde{f_i}\|$ (see line 93 - 99 in our manuscript), which is equivalent to the norm of the (S)MoE output. Fig. 2 verifies this expectation by showing that each MoE and SMoE layer learns to reduce its output norm during training, suggesting that the scores $\text{softmax}(g_i(x_t)) $ and $\text{softmax}(\text{TopK}(g_{i}(x_t)))$ are learned to approximate $\alpha_i^{*}$. These explain line 140.
>
> **Q3. Grammar issues:** We have addressed the grammar issues in our revision.
>
> **Q4. Insufficient discussion of computation overhead.**
>
> **Answer:** The reviewer might have overlooked Appendix E.5 of our manuscript. Table 11 in E.5 reports the training/inference runtime, training/inference memory usage, and the number of parameters of our MomentumSMoE/AdamSMoE vs. the SMoE baseline. Our MomentumSMoE/AdamSMoE have the same training/inference memory usage and number of parameters as SMoE. Furthermore, our momentum models are on par with the SMoE baseline in terms of training/inference runtime. Compared to the SMoE baseline, our MomentumSMoE (see Fig. 1) only needs an additional momentum state $p$, which is accumulated from layer to layer. Thus, MomentumSMoE only uses an additional tensor shared between layers to store this momentum state $p$. Similarly, AdamSMoE only needs two additional tensors shared between layers to store $p$ and $m$. The memory overhead for introducing these additional tensors is minimal, thus explaining why memory increase and throughput decrease in our MomentumSMoE/AdamSMoE compared to the SMoE baseline are negligible.

---

> ### Author Response · Authors · 2024-08-10
> **Any Questions from Reviewer oJG3 on Our Rebuttal?**
>
> We would like to thank the reviewer again for your thoughtful reviews and valuable feedback.
>
> We would appreciate it if you could let us know if our responses have addressed your concerns and whether you still have any other questions about our rebuttal.
>
> We would be happy to do any follow-up discussion or address any additional comments.

---

> > ### Comment · Reviewer_oJG3 · 2024-08-10
> > **After Rebuttal**
> >
> > Thanks to the detailed response from authors. Most of my concerns are well addressed. I'd like to increase the score to 5.

---

> > > ### Author Response · Authors · 2024-08-10
> > > **Thanks for your endorsement!**
> > >
> > > Thanks for your response, and we appreciate your endorsement.

---

### Official Review · Reviewer_qEGD · 2024-07-17

**Soundness:** 3
**Presentation:** 3
**Contribution:** 3
**Rating:** 6
**Confidence:** 4

**Summary:**

This paper addresses the instability problem of training SMoE models. By establishing a relationship between SMoE and multi-objective optimization, the authors integrate momentum into SMoE and propose MomentumSMoE. Experimental results show that MomentumSMoE is more stable than SMoE during training.

**Strengths:**

1. The paper tackles a critical issue in the training of SMoE models.

2. The proposed method is generalizable and can be applied to various SMoE models such as V-MoE and GLaM.

3. Experimental results demonstrate that this method is more stable than SMoE during the training process.

**Weaknesses:**

1. This method has little effect on models with few layers.

2. The largest models for evaluation only have  388M parameters, which are much smaller than mainstream MoE LLMs.

3. From a theoretical standpoint, developing a framework to explain the enhanced robustness of MomentumSMoE would be interesting.

**Questions:**

please refer to weaknesses

**Limitations:**

The authors have adequately discussed the limitations.

---

> ### Author Rebuttal · Authors · 2024-08-06
>
> **Q1. This method has little effect on models with few layers.**
>
> **Answer:** A momentum-based approach like MomentumSMoE needs more layers to show its advantages, just like a heavy-ball momentum, or Adam needs a couple of iterations to start showing its faster convergence compared to gradient descent. Also, models with few layers are not common in practice due to their worse performance compared to models with more layers. For example, as can be seen in Table 7 in our manuscript, the SMoE-small baseline has much lower validation and test perplexity (PPL) compared to SMoE-medium and large baselines, 84.26/84.81 PPL vs. 33.76/35.55 and 29.31/30.33 PPL, respectively. The poor PPL of the SMoE-small baseline renders its performance on the WikiText-103 language modeling tasks negligible.
>
> **Q2. The largest models for evaluation only have 388M parameters, which are much smaller than mainstream MoE LLMs.**
>
> **Answer:** Thanks for your comment. During the rebuttal, we have tried our best to conduct an additional experiment on a Sparse Mixture of Experts (SMoE) backbone with more than 1B parameters. However, due to the shortage of computing resources and the short time of the rebuttal, our trainings have not finished yet. We will include these results of our momentum-based SMoE on a larger SMoE backbone in the revised manuscript and during the discussion period if possible.
>
> **Q3. From a theoretical standpoint, developing a framework to explain the enhanced robustness of MomentumSMoE would be interesting.**
>
> **Answer:** Thanks for your suggestion. We agree with the reviewer that developing a framework to explain the enhanced robustness of MomentumSMoE would be interesting. The approach to theoretically proving the robustness of MomentumSMoE is to consider a minimizer $x^*$ of the objective function $F(x)$ in Eqn. (3) in our manuscript, which is found by heavy-ball updates when starting at the initial point $x_0$. Then, we prove that by tuning the momentum $\mu$ and step size $\gamma$, the heavy-ball iterations will find a point $x_k$ such that $\||x_k - x^*\|| \le c\rho^{k}$ for a constant c, $\rho \in [0, 1)$, and $k \ge 1$. This result implies that the output of MomentumSMoE is close to $x^{*}$ as long as the input data is close to $x_0$, thus verifying the robustness of MomentumSMoE. This proof can be extended from the proof of Proposition 1 and Corollary 1 in our manuscript. We will include the detailed proposition and its proof for the robustness of MomentumSMoE in our revision.

---

> ### Author Response · Authors · 2024-08-10
> **Any Questions from Reviewer qEGD on Our Rebuttal?**
>
> We would like to thank the reviewer again for your thoughtful reviews and valuable feedback.
>
> We would appreciate it if you could let us know if our responses have addressed your concerns and whether you still have any other questions about our rebuttal.
>
> We would be happy to do any follow-up discussion or address any additional comments.

---

### Author Rebuttal · Authors · 2024-08-07

## Global Rebuttal

Dear AC and reviewers,

Thanks for your thoughtful reviews and valuable comments, which have helped us improve the paper significantly. We are encouraged by the endorsements that: 1) The idea of integrating momentum into Sparse Mixture of Experts (SMoE) is original, interesting, a non-trivial innovation that tackles a critical issue in the training of SMoE models (all reviewers); 2) The paper provides convincing and comprehensive empirical evidence showing the effectiveness of MomentumSMoE across multiple benchmarks (all reviewers); 3) The proposed method is compatible with other momentum-based optimizers, like Adam (Reviewer oJG3), and can be applied to various SMoE models such as V-MoE and GLaM (Reviewer qEGD). We have included the additional experimental results requested by the reviewers in the 1-page attached PDF.

One of the concerns shared by Reviewer oJG3 and GFpG is that the formulation of SMoE as a multi-objective optimization problem is not clear and the connection between SMoE and gradient descent needs justification. We address this concern here.

**Clarifying the formulation of SMoE as a multi-objective optimization (MOO) problem**

Let us start by rewriting the minimization problem in Eqn. (3) as follows:

$$
\min_{x\in D, \theta \in \Theta} F(x, \theta) := \sum_{i=1}^E c_i F_i(x, \theta^{i}).
$$
where $\theta^{i}$ is the parameters of the expert network $i$, $\theta = \{\theta^{1},\dots,\theta^{E}\}$, and $\Theta$ is the parameter space. Also, $D$ is the feasible region, and $c_i \in \mathbb{R}$, $i=1,\dots,E$, are weights representing the importance of each objective function. In order to solve this optimization problem, we employ the alternating minimization approach as follows:

*Step 1 - fix $\theta$ and minimize $F$ w.r.t. $x$*:

$$
x_{t+1} = x_{t} - \gamma \sum_{i=1}^{E}\alpha_{i}^*\nabla_xF_i(x_t,\theta^{i}_{t})
$$

$$
= x_{t} - \gamma \sum_{i=1}^{E}\alpha_{i}^* f_{i}(x_{t}, \theta_{t}^i),
$$

where $\alpha^*=(\alpha_1^*,\dots,\alpha_E^*)$ satisfy the Pareto-stationary condition in Definition 1 in our manuscript, $\gamma$ is the step size, and $f_{i}=\nabla_xF_i$.

*Step 2 - fix $x$ and minimize $F$ with respect to $\theta$*
$$
\theta_{t+1}^{i} = \min_{\theta^{i} \in \Theta^{i}}  c_i F_i(x_{t+1}, \theta^{i}), i=1,\dots,E.
$$
where $\theta^{i}$ lies in the parameter space $\Theta^{i}$.

In our formulation of SMoE as a multi-objective optimization problem, we regard $-u_i(x_t)$, where $u_i$ is the $i^{\text{th}}$ expert network,  as the gradient $\nabla_{x}F_i(x_t,\theta^{i}_{t})$ in Step 1, and the score $\text{softmax}(g(x_t))_i$ from the router $g(x_t)$ is learned to approximate $\alpha_i^*$. Step 1 in the alternating minimization algorithm above corresponds to the forward pass of an SMoE, and step 2 corresponds to an update step of the model's parameters (i.e., similar to an M-step in an EM algorithm). Note that this parameter update step is implicitly performed at each training iteration via optimizing the objective loss at the final layer of the model using backpropagation and gradient descent. Results in Fig 2. and the discussion from line 137 to line 144 in our manuscript provide supporting evidences for this implicit parameter update.

As can be seen in Step 1, the gradient $\nabla_{x}F_i(x_t,\theta_{t}^{i})$ is a function of $\theta_{t}^{i}$ and is supposed to change during the model training, which matches the observation that expert network is continually changing during the model training. Also, the objective function of our MOO problem is $F(x, \theta)$ in which we want to find $x^*$ and $\theta^*$ to minimize $F$. This objective function is static, clear, and stable.

**Empirical evidences to justify the connection between the SMoE and gradient descent**

In Section 2.3 of our manuscript, from line 128 to line 144, we discuss the empirical evidences (see Figure 2 and 3 in our manuscript) for connection between SMoE and gradient descent. In particular, from Eqn. (8) and (9), the output of (S)MoE corresponds to $-f(x_t)$, where $f(x_t) = \nabla_xF(x_t)$ and is the gradient of the objective function $F$ with respect to $x$ at $x_t$. If (S)MoE indeed performs (stochastic) gradient descent in its forward pass, then the norm of the (S)MoE output must decrease when t (i.e., layer index) increases since the gradient norm $\||f(x_t)\||$ decreases when gradient descent updates are applied. Fig. 3 confirms this by showing that the norm of the (S)MoE output decreases over layers in a 6-layer (S)MoE model trained on the WikiText-103 language modeling task. At the last layer, the norm increase might be due to overshooting, a common phenomenon that can occur when using gradient descent.

Additionally, from line 116 to line 119 in our manuscript, we hypothesize that the scores $\text{softmax}(g_i(x_t))$ for MoE and $\text{softmax}(\text{TopK}(g_{i}(x_t)))$ for SMoE from the router $g(x_t)$ are learned to approximate $\alpha_i^*$. If this is indeed true, then in order to satisfy the Pareto-stationary condition in Definition 1 in our manuscript, the scores should be learned to minimize the norm $\||\sum_{i=1}^E \alpha_i \tilde{f_i}\||$ (see line 93 - 99 in our manuscript), which is equivalent to the norm of the (S)MoE output. Fig. 2 verifies this expectation by showing that each MoE and SMoE layer learns to reduce its output norm during training, suggesting that the scores $\text{softmax}(g_{i}(x_t))$ and $\text{softmax}(\text{TopK}(g_{i}(x_t)))$ are learned to approximate $\alpha_i^{*}$.

-----

We hope that our rebuttal have cleared your concerns about our work. We are glad to answer any further questions you have on our submission, and we would appreciate it if we can get your further feedback at your earliest convenience.

---

### Decision · Program_Chairs · 2024-09-25

**Decision:**

Accept (poster)

**Comment:**

This paper introduces several momentum-based algorithms designed to enhance the stability and robustness of Sparse Mixture of Experts (SMoE). As many reviewers have noted, this approach is novel and holds significant potential for impact within the community. The paper effectively demonstrates both theoretical and empirical improvements of MomentumSMoE over standard SMoE across various tasks.

After personally reading the paper, I observed that MomentumSMoE aligns more closely with fixed-point iteration acceleration techniques rather than traditional momentum-based gradient descent. Additionally, as discussed in Section 3, the assumptions that the sequence $x_0, \dots, x_t $ lies in the neighborhood of the optimal point $ x^*$, and that $ x^* = 0$, are not convincingly justified in practical scenarios. Furthermore, as Reviewer GFgG pointed out, a detailed analysis of the spectrum of $\nabla_x f(x)$ would provide a deeper understanding of the effectiveness of MomentumSMoE.

Despite these concerns, the novelty and potential impact of the proposed methods are significant. Therefore, I  recommend accepting this paper.